# Efficient and Privacy-Preserving Soft Prompt Transfer for LLMs

Xun Wang [1]   Jing Xu [1]   Franziska Boenisch [1]   Michael Backes [1]   Christopher A. Choquette-Choo [2]
Adam Dziedzic [1]

## Abstract

Prompting has become a dominant paradigm for adapting large language models (LLMs). While discrete (textual) prompts are widely used for their interpretability, soft (parameter) prompts have recently gained traction in APIs. This is because they can encode information from more training samples while minimizing the user's token usage, leaving more space in the context window for task-specific input. However, soft prompts are tightly coupled to the LLM they are tuned on, limiting their generalization to other LLMs. This constraint is particularly problematic for *efficiency* and *privacy*: (1) tuning prompts on each LLM incurs high computational costs, especially as LLMs continue to grow in size. Additionally, (2) when the LLM is hosted externally, soft prompt tuning often requires sharing private data with the LLM provider. For instance, this is the case with the NVIDIA NeMo API. To address these issues, we propose POST (**P**rivacy **O**f **S**oft-prompt **T**ransfer), a framework that enables private tuning of soft prompts on a small model and subsequently transfers these prompts to a larger LLM. POST uses knowledge distillation to derive a small model directly from the large LLM to improve prompt transferability, tunes the soft prompt locally—optionally with differential privacy guarantees—and transfers it back to the larger LLM using a small public dataset. Our experiments show that POST reduces computational costs, preserves privacy, and effectively transfers high-utility soft prompts. Our code is available at `https://github.com/sprintml/POST`.

## 1. Introduction

Large Language Models (LLMs) are strong general-purpose language generators that can be adapted to solve various private downstream tasks (OpenAI, 2023; Gemini-Team et al., 2023). One prominent paradigm for adapting LLMs to private tasks is prompting (Devlin et al., 2018; Radford et al., 2018). Prompting is a more flexible alternative to fine-tuning, as it allows multiple tasks to be solved simultaneously in a single inference pass, leading to higher efficiency and reducing the need for storing multiple fine-tuned model copies. While *discrete prompts* (Schick & Schütze, 2020; 2021a; Shin et al., 2020; Han et al., 2022), which prepend textual tokens to the LLM's input, have been shown relatively successful for LLM adaptations, they require large engineering efforts and lots of trials and errors. As an alternative, *soft prompts* (Shin et al., 2020; Lester et al., 2021; Li & Liang, 2021; Zhong et al., 2021; Oymak et al., 2023) prepend trainable embedding vectors to the input, which can be tuned automatically on the private downstream data using standard gradient-based approaches. Such gradient-based approaches are generally known to yield higher performance at lower computational costs (Liu et al., 2022). Additionally, soft prompts can encode information from many more private downstream examples while using fewer input tokens than discrete prompts. This reduces costs—since many APIs charge per token—and leaves more of the context window for the downstream task (Varshney & Surla, 2023).

Yet, soft prompt tuning has two major limitations. 1) As LLMs grow in size (Geng & Liu, 2023; Chiang et al., 2023; Brown et al., 2020), it requires significantly more computation to backpropagate through the entire LLM. 2) Backpropagation requires the model and data to be *on the same device*. For centrally hosted LLMs, this leaves two options: either the user needs to share their private data with the LLM provider, or the LLM provider needs to share their LLM with the user. The latter is impractical as users usually do not have the compute to deploy an LLM. Additionally, the model is the provider's intellectual property (Dziedzic et al., 2022a;b;c; Dubiński et al., 2023) and should be protected. Thus, sharing it would, at the very least, disrupt the provider's business model, as users would no longer be required to pay per query. Therefore, the former, *i.e.,* users sharing their data with the LLM provider, is implemented

---

[1]CISPA Helmholtz Center for Information Security, Saarbrücken, Germany [2]Google DeepMind. Correspondence to: Franziska Boenisch <boenisch@cispa.de>, Adam Dziedzic <adam.dziedzic@cispa.de>.

*Proceedings of the 42$^{nd}$ International Conference on Machine Learning*, Vancouver, Canada. PMLR 267, 2025. Copyright 2025 by the author(s).

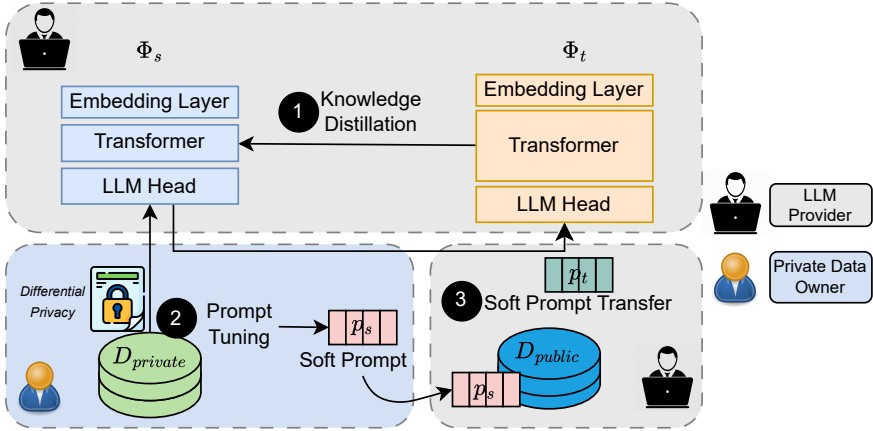

Figure 1: **POST**⊠ **Framework.** ❶ An LLM provider compresses their LLM $\Phi_t$ into a smaller model $\Phi_s$ through knowledge distillation. ❷ The private data owner learns a specific soft prompt $p_s$ on $\Phi_s$ using their private dataset (optionally with differential privacy guarantees). ❸ The LLM provider obtains the soft prompt $p_t$ for solving the user's task by transferring $p_s$ to the target LLM $\Phi_t$—solely relying on a small public dataset and *no access to the private data for transfer.*

in popular APIs, such as NVIDIA's Nemo API (Varshney & Surla, 2023). However, this approach causes severe privacy concerns for the users when the data is private or sensitive in nature (Duan et al., 2023a;b; Hanke et al., 2024).

A potential solution to both problems is to tune the soft prompt locally on a smaller model and then transfer and use it on the large LLM. This approach, commonly known as "prompt transfer" (Su et al., 2022; Wu et al., 2023; Xiao et al., 2023), has proven effective for discrete prompts (Rakotonirina et al., 2023; Hong et al., 2023; Wen et al., 2023). However, soft prompts are highly coupled to the LLM they were tuned on, making them difficult to transfer. Therefore, existing attempts to transfer soft prompts between LLMs face severe limitations: either they require private data access for the LLM provider during transfer (Su et al., 2022), which—as we discussed—raises privacy concerns, or they are ineffective, as the transferred prompts' utility on the large central LLM often underperforms even compared to the prompted small model (Wu et al., 2023), disincentivizing the transfer altogether.

Our proposed **POST** (**P**rivacy **O**f **S**oft-prompt **T**ransfer) framework overcomes these limitations. POST consists of three key steps as shown in Figure 1. (1) The LLM provider performs *knowledge distillation* (Hinton et al., 2015) to compress their LLM into a smaller language model. Note that this is a *one-time* operation, and the same small model can be used by any user for their various tasks. After distillation, the LLM provider sends the small model to the user. Then, (2) the user performs *local prompt tuning* using their private data on this smaller model, potentially incorporating formal privacy guarantees through differential privacy (Dwork et al.,

2006). Once the user has tuned the private prompt on their data, they provide this prompt to the LLM provider, who (3) *transfers the prompt* to achieve strong performance on the large LLM. To perform this transfer without any additional leakage from the user's private data to the LLM provider, we equip POST with a novel prompt transfer method that relies purely on access to a small public dataset during transfer rather than the user's private data.

Our thorough experimental evaluation on both masked language models and auto-regressive language models demonstrates that our method can efficiently, effectively, and privately transfer soft prompts with high utility. Thus, POST has the potential to enable more LLM providers to integrate soft prompts into their APIs, unlocking benefits such as multi-task inference, improved adaptability, extended context windows for users—all while maintaining user privacy.

In summary, we make the following contributions:

- We propose POST, a framework for privacy of soft prompt transfer. POST preserves the confidentiality of users' private data and can also provide strong privacy guarantees through differential privacy.

- We design a novel method to transfer private prompts between LLMs by purely relying on public data which we integrate into POST.

- We provide detailed experimental analysis using five classification datasets and two open-ended generation tasks to simulate our setup and three different types of LLMs to show the effectiveness and efficiency of our method.

## 2. Background

**Prompt Tuning** (PT) aims at adapting a publicly pre-trained LLM to various natural language downstream tasks. There are two major types of prompts, 1) *hard or discrete prompts* (Schick & Schütze, 2021a;b; Gao et al., 2021), which are discrete textual tokens prepended to the input text of the LLM, and 2) *soft prompts* (Hambardzumyan et al., 2021; Qin & Eisner, 2021; Zhong et al., 2021), which are tunable embedding vectors prepended to the LLM's input. While discrete prompts require thorough engineering to yield good performance on downstream tasks, soft prompts can be tuned through standard gradient-based training approaches (Lester et al., 2021). Formally, given an input sequence with $n$ tokens $X = \{x_1, x_2, \ldots, x_n\}$, labeled by $y$, we first prepend $l$ randomly initialized soft prompts $P = \{\mathbf{p}_1, \mathbf{p}_2, \ldots, \mathbf{p}_l\}$ before $X$, where $\mathbf{p}_i \in \mathbb{R}^d$ is an embedding vector, and $d$ is the input dimension of the LLM. The training objective is to maximize the likelihood of decoding the correct output $y$ as $\mathcal{L} = p(y|P, X)$. The key is that the model itself remains frozen and only $P$ is tunable.

**Knowledge Distillation** (KD) (Hinton et al., 2015; Buciluă et al., 2006) is a compression method for machine learning models. It works by transferring knowledge from a complex model, denoted as the *teacher model*, to a simpler, smaller model known as the *student model*. KD has been shown effective in compressing LLMs during the pre-training phase while maintaining their performance (Sanh et al., 2019; Gu et al., 2024; Sreenivas et al., 2024). Already pre-trained LLMs can also be compressed successfully through KD (Gu et al., 2024; Wang et al., 2025). In prompt transfer, Zhong et al. (2024) leverage knowledge distillation to alleviate forgetting between tasks that use transferred prompts. In contrast, our setup considers transferring prompts between models. While, in general, most of the KD approaches aim at generating a student with a similar predictive performance as the teacher, for our purpose, the student's performance is not particularly relevant. The student just needs to match the teacher's predictive behavior to a certain degree in order to facilitate the transfer of the prompt from student to teacher.

**Soft Prompt Transfer** can be computationally expensive, especially as LLMs grow in size, because soft prompts are trained with backpropagation through the entire LLMs. This motivates the emergence of attempts to transfer, *i.e.,* to reuse, existing soft prompts. There are two broad scenarios for prompt transfer. The first one aims at reusing a soft prompt trained for one downstream task on another (similar) downstream task on the same LLM (*cross-task transfer)*. This can be implemented, for example, by initializing the parameters of the second soft prompt with the trained existing soft prompt parameters and has been shown to reduce training time for the second prompt (Vu et al., 2022; Su et al., 2022; Zhong et al., 2024; Philippy et al., 2024). The second and

more challenging scenario for prompt transfer is a *cross-model transfer* scenario. In this scenario, one tries to tune a prompt for a given task on one LLM, and then use it for another LLM. The difficulty arises from the fact that soft prompts (over)fit the LLM they were tuned for and usually do not exhibit a strong performance on other LLMs. Su et al. (2022) address transferring the soft prompt between the LLMs by using the guidance of the private data. However, this approach still exposes the data directly to the second LLM which may not be possible when this LLM is hosted centrally by a service provider (*e.g.,* OpenAI) and the data is sensitive in nature, as the private data now needs to be shared with the external provider. Wu et al. (2023) present a zero-shot prompt transfer method, where source prompts tuned on a given LLM are encoded into a relative space and used as a form of support vector when finding target prompts on the second, *i.e.,* target model. Unfortunately, in their approach, the target model with the transferred prompt performs worse than the source model, leaving no incentive to use the target model rather than the source model with the prompt. In contrast, our method significantly improves performance on the target models with the transferred prompts. Additionally, their transfer requires the private data, which thereby leaks entirely to the model owner. In contrast, our method performs prompt transfer *solely with public data*, preserving confidentiality of the private data towards the model provider. Transferring tasks between LLMs has also been explored by Xiao et al. (2023) for transfer learning. While they focus on fine-tuning and their approach is not applicable for soft-prompt tuning, we operate in the same setup and under the same assumptions as they do.

**Differential Privacy** (DP) (Dwork, 2006) is a mathematical framework that provides privacy guarantees for ML by implementing the intuition that a model $\mathcal{M} : I \to S$, trained on two neighboring datasets $D$, $D'$ that differ in only one data point, will yield roughly the same output, *i.e.,* $\Pr[\mathcal{M}(D) \in S] \leq e^\varepsilon \cdot \Pr[\mathcal{M}(D') \in S] + \delta$. The privacy parameter $\varepsilon$ specifies by how much the output is allowed to differ, and $\delta$ is the probability of failure to meet that guarantee. To adapt soft prompts under differential privacy (DP) guarantees, Duan et al. (2023a) introduced the **PromptDPSGD** algorithm. Initially developed for text classification, PromptDPSGD was subsequently adapted to support text generation tasks as well (Hanke et al., 2024). The algorithm builds on the widely used Differentially Private Stochastic Gradient Descent (DPSGD) method (Abadi et al., 2016), in which each gradient is clipped to bound its sensitivity (to limit how much each data point influences the trained parameters), and calibrated Gaussian noise is added to the aggregated gradients to ensure a finite DP guarantee.

**Private Prompting and Text-to-Text Privatization** There exist multiple DP frameworks for private prompting (Duan et al., 2023b; Tang et al., 2024; Wu et al., 2024). However,

they mainly operate in a different setup and only provide DP guarantees for the model *output*, yet leak the private data to the model provider (Hanke et al., 2024). In contrast, our work aims at protecting the private data *against the model provider*. In a similar vein to our work, **DP-OPT** (Hong et al., 2023) tries to avoid leakage to the model owner and tunes discrete prompts with DP guarantees on a local surrogate LLM and then transfers these prompts to the large LLM. Their focus on discrete prompts (in contrast to our work that relies on soft prompts) leads to certain words and phrases from the training dataset leaking directly to the LLM provider, as shown in their Figure 3—which does not occur with our method. Additionally, their results show that the local surrogate model needs to be of strong performance (*i.e.,* large) to obtain good transfer results, leading to very high compute requirements on the user side. In contrast, our method relies on small surrogate models that can be used for prompt tuning with low computing on the user's end. We empirically compare against their method in Table 5.

## 3. Setup and Problem Formulation

**The Setup.** We consider two parties: an LLM provider and a user. The LLM provider deploys a general-purpose LLM and offers paid query access to it. The user holds private data and wants to adapt the LLM on this data to solve their downstream tasks while ensuring the confidentiality and privacy of the data towards the LLM provider.

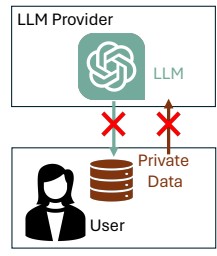

**The Problem.** Unfortunately, for soft prompt tuning—which involves gradient calculation—both the data and the LLM are required to be on the same device. The problem is that the user may not be able to share their data with the LLM provider due to privacy concerns while the LLM provider cannot share their LLM because of 1) intellectual property concerns and since 2) this would disrupt their business model, as users would no longer be required to pay for accessing model queries. Additionally, most users would lack the necessary computational resources to tune the soft prompt on the large LLM locally, as this requires calculating gradients over the entire model. Due to these limitations, the powerful LLM cannot be used for private tasks.

**Our Goal and Solution.** Our goal is to protect the private downstream data (*i.e.,* the "training data") used to tune the soft prompt from leaking to the LLM provider.[1] We propose a solution based on tuning the soft prompt on a small local model and then transferring this prompt to the LLM by using

public data. To obtain a suitable small model that facilitates prompt transfer, we propose that the LLM provider performs KD from their LLM. The small model needs to meet three critical requirements: it must (i) be small enough to enable the user to perform local soft prompt tuning on their own hardware, (ii) closely match the semantics of the original LLM to facilitate an effective prompt transfer, and, from the perspective of the LLM provider, (iii) be limited in performance to ensure that users still have an incentive to use the original LLM rather than the small prompted version. Note that the KD is a *one-time* operation for the LLM provider as the same small model can be used by any user for their various tasks. After distillation, the small model is sent to the user who tunes a soft prompt on it using their private data, potentially with DP to formally bound privacy leakage. Finally, the tuned prompt is sent to the LLM provider who performs a prompt transfer step relying on public data. Eventually, the client can use the LLM using the transferred prompt. We provide an overview of this solution in Figure 1 and detail its building blocks in the following section.

## 4. Our Private Transfer of Soft Prompts Framework

Our **P**rivacy **O**f **S**oft-prompt **T**ransfer (POST) framework consists of three main building blocks: (1) a KD from the LLM to a small model, (2) private prompt tuning, and (3) a privacy-preserving prompt transfer using public data. We detail those building blocks below.

### 4.1. Knowledge Distillation

To derive a small model $\Phi_s$ from the large LLM $\Phi_t$, we rely on KD. We discuss the advantage of KD for our POST framework over other alternatives for model compression, such as quantization and pruning, in Appendix D.2.

We follow the approach of Sanh et al. (2019) to derive $\Phi_s$ from $\Phi_t$. Their distillation objective function is given by

$$\mathcal{L}_{distil} = \alpha_{ce}\mathcal{L}_{ce} + \alpha_{lm}\mathcal{L}_{lm} + \alpha_{cos}\mathcal{L}_{cos}. \quad (1)$$

This objective combines three losses: distillation loss $\mathcal{L}_{ce}$, language modeling loss $\mathcal{L}_{lm}$, and embedding cosine loss $\mathcal{L}_{cos}$. The distillation loss $\mathcal{L}_{ce}$ measures the Kullback-Leibler divergence between the logits of $\Phi_s$ and $\Phi_t$. The language modeling loss $\mathcal{L}_{lm}$ corresponds to the standard pretraining objective, using cross-entropy to predict masked or next tokens. The embedding cosine loss $\mathcal{L}_{cos}$ computes the cosine distance between the embeddings of $\Phi_s$ and $\Phi_t$. These three losses are weighted by $\alpha_{ce}$, $\alpha_{lm}$, and $\alpha_{cos}$.

In contrast to prior work (Sanh et al., 2019; Xiao et al., 2023) that relies on KD to moderately compress LLMs while maintaining high performance on $\Phi_s$, our goal is to achieve

---

[1]Protecting the queries (*i.e.,* the "test data") from leaking to the LLM provider, often referred to as *private inference*, is orthogonal and out of scope for this work.

a *significant compression* without focus on maintaining $\Phi_s$'s performance. Thereby, we ensure that $\Phi_s$ is small enough to fit into the user's hardware for local prompt tuning and is not performant enough to disincentivise the (paid) use of $\Phi_t$, as desired by the LLM provider. We perform extensive ablations on the setup for the KD, the importance of the individual loss term in Equation (1), and the resulting trade-offs between the compressed model's performance and our POST's success in Section 5.6.

## 4.2. Private Prompt Tuning

The goal is to tune a local prompt $p_s$ on the small source model $\Phi_s$ using the private data $D_{pri}$ such that $p_s$ minimizes the loss $\mathcal{L}$ on the private downstream task as

$$\underset{p_s}{\arg\min} \sum_{x \in D_{pri}} \mathcal{L}(\Phi_s, p_s + x), \quad (2)$$

where $x$ denotes a query input sequence from the task that we want to predict. This approach can be performed with standard PT. It provides confidentiality for the private data since the data is not directly sent to the LLM provider. Recent work (Duan et al., 2023b), however, highlights that private information can leak from tuned discrete prompts. In Section 5.4, we show that this also holds for soft prompts, motivating the necessity of integrating privacy guarantees.

To formally bound privacy leakage, we propose to tune $p_s$ with DP, for example, using the PromptDPSGD algorithm (Duan et al., 2023a). During optimization, Prompt-DPSGD clips the per-sample gradients of the loss to a clip norm $c$ and adds Gaussian noise drawn from $\mathcal{N}(0, \sigma^2 c^2 I)$ to provide $(\varepsilon, \delta)$-DP guarantees (see Appendix C.3 in Duan et al. (2023a) for the privacy proof). Hence, applying their algorithm bounds the leakage of the private prompt tuning data $D_{pri}$ from $p_s$ to $(\varepsilon, \delta)$. Because of the DP post-processing guarantees, $p_s$ can then be shared with the LLM provider without incurring further privacy loss.

## 4.3. Privacy-Preserving Prompt Transfer through Public Data

The prompt $p_s$, tuned on the small source model $\Phi_s$, could, in principle, be directly applied to the large target LLM $\Phi_t$. However, as described above, soft prompts fit very strongly the model that they were tuned on. Hence, they do not initially perform very well on other LLMs. A naive solution is to fine-tune the target prompt $p_t$ on the private data $D_{pri}$. However, this would disclose the private data to the LLM provider and is, hence, not acceptable. As an alternative, we propose the first privacy-preserving prompt transfer that leverages a small *public* dataset $D_{pub}$ in an efficient transfer step to derive a high-utility prompt $p_t$ from $p_s$.

We start by initializing the target prompt $p_t$ with the same initialization of $p_s$, then iteratively update $p_t$. For the itera-

tive update, we design a loss function

$$\mathcal{L} = (1 - \alpha)\mathcal{L}_1 + \alpha\mathcal{L}_2, \quad (3)$$

that consists of two different loss terms. The first loss term aims at mimicking the prompted small model's predictions on the public data closely. It is defined as

$$\mathcal{L}_1 = \sum_{\hat{x} \in D_{pub}} \text{KLDiv}(\Phi_t(p_t + \hat{x}), \Phi_s(p_s + \hat{x})), \quad (4)$$

where KLDiv denotes the Kullback–Leibler divergence. The second loss term aims at replicating the changes to the initial model behavior introduced by the prompt. Formally, the second loss aligns the direction change induced by the private prompt between $\Phi_t$ and $\Phi_s$, on the public data as

$$\mathcal{L}_2 = \sum_{\hat{x} \in D_{pub}} \text{KLDiv}((\Phi_t(p_t+\hat{x})-\Phi_t(\hat{x})), (\Phi_s(p_s+\hat{x})-\Phi_s(\hat{x})).$$
$$(5)$$

We need both loss terms, since, particularly when the small model does not have good performance, just mimicking its behavior, rather than matching the impact of the prompt, is not effective.

Based on the design of our loss function, the best choice for $\alpha$ in Equation (3)—controlling the balance between the two loss terms—depends on the zero-shot accuracy of the large LLM on the private task and the performance of the compressed LLM. If the large LLM has a good zero-shot performance, or the compressed LLM has a weak performance, a larger $\alpha$, *i.e.,* putting the emphasis $\mathcal{L}_2$ to mimic the prompt behavior, rather than improving the task performance, is more beneficial. We present extensive ablations and guidelines on the choice of $\alpha$ in Section 5.6.

## 5. Empirical Evaluation

### 5.1. Experimental Setup

**Models and Distillation.** We perform experiments with *three LLMs* of diverse scale, namely Roberta-base (Liu et al., 2019), GPT2-XL (Radford et al., 2019), and Llama2-7b (Touvron et al., 2023). We experiment with various degrees of compression in the KD (see Appendix D.4). For the results presented in the main body of the paper, we compress the 12-layer Roberta-base and the 32-layer Llama2-7b to 2 layers, and the 48-layer GPT2-XL to 4 layers. We use the Bookcorpus (Zhu et al., 2015) dataset for the KD.

**Datasets.** Following prior work (Hong et al., 2023; Wu et al., 2023), we evaluate the performance of our proposed method on *five classification-task datasets*: sst2 from the GLUE benchmark (Wang et al., 2019), imdb (Maas et al., 2011), tweet (Rosenthal et al., 2017), arisetv (Okite, 2022) and mpqa (Wiebe et al., 2005). To show the general applicability of our method on *open-ended generation tasks*,

we go beyond what was shown in prior work, and additionally perform evaluations on the MIT dataset (Liu et al., 2012), which aims to extract the genre or director of a movie given the movie description. This task requires the generation of multiple tokens with varying lengths, highlighting our POST's flexibility. As public data, we also include agnews (Zhang et al., 2015) and boolq (Clark et al., 2019) for the classification task, while for the generation task, we use AIE (Kudari, 2022). We also include disaster (CrowdFlower, 2019) and trec (Li & Roth, 2002) for baseline comparison and ablation on the choice of public data. Further details on the datasets and their tasks are provided in Appendix B.2.

**KD, Prompt Tuning, and Prompt Transfer.** We experiment with various setups for the KD. If not indicated otherwise, we use the KD hyperparameters from Sanh et al. (2019). See Appendix B.1 for more details. Following Su et al. (2022), we initialize our soft prompts with 100 tokens. We include an ablation study on prompt length in Section 5.6. The hyperparameters for prompt tuning per dataset, including the $\delta$ for the DP setup, are presented in Table 10. During the prompt transfer, the model provider has no access to the private dataset to find the right moment to stop the transfer, so we report the transferred accuracy after a fixed number of steps. We experiment with various numbers of steps and public data points used for transfer in Section 5.6. By default, we use 5000 steps for Roberta-base, 8000 steps for GPT2-XL, and 6000 steps for Llama2-7b.

**Metrics and Baselines.** To evaluate the success of our method, we report the accuracy on the test data split of our private datasets for the teacher LLM with the transferred prompt (**Private Transfer**). As baselines for comparison, we include the zero-shot performance of the teacher LLM on the private tasks' test sets (**Full ZS**), representing the lower bound our method should improve upon. Additionally, we provide the performance of tuning the prompt for the teacher LLM on the private training data, which, due to privacy concerns, is not feasible in practice (**Full PT**). This serves as the theoretical upper bound for potential performance. We also report the accuracy of the prompted compressed model after tuning the prompt on it (**Compressed PT**), as our private transfer must improve over this metric to justify using the teacher LLM instead of the small student model. Finally, we report the direct transfer accuracy (**Direct Transfer**), which is the accuracy achieved when the prompt tuned on the small model is directly applied to the large one. The improvement of POST over this value shows the effectiveness of our prompt transfer step.

### 5.2. Effective Prompt Transfer with POST

As the main result, in Table 1, we show POST's effectiveness in transferring soft prompts in both setups, with and without DP. For each private dataset, we experiment with

two different public datasets for prompt transfer and report the respective transferred accuracy on the private dataset. We first observe that the transferred performance is significantly higher than the zero-shot performance of the large LLM (Full ZS). This highlights the improvements that a user can gain from adapting the large LLM to their private task using our method. Additionally, after prompt transfer with POST, we outperform the small compressed prompted model (Compressed PT), giving users a strong incentive to transfer their prompt back to the (paid) large LLM, rather than using the small model locally. Further, we show that our prompt transfer described in Section 4.3 is highly effective as it improves over the direct transfer (Direct Transfer) performance by a large margin. In contrast to the soft prompt transfer method by Wu et al. (2023) which showed a *decrease* in accuracy after transfer, our results highlight for the first time the practical applicability and the benefits of transferring soft prompts.

When using DP during the prompt tuning, we observe that the improvement of the transfer performance to the large LLM over the performance on the prompted compressed model (Compressed PT) is even more significant than in the non-DP setup. For example, on the sst2 dataset, using imdb as public data, we observe an improvement of 18.92% for the DP case, while we only have an improvement of 11.24% in the non-DP case (see first lines of Table 1b and Table 1a, respectively). We hypothesize that the noise added for DP during tuning acts as a regularizer that can help to prevent overfitting on the small sensitive datasets and the distilled model, hence, generalizing better to the large LLM. Overall, our results are very close to the non-private upper bound baseline (Full PT) of tuning the prompt directly on the large LLM, which would not only leak private information from the model outputs to third parties querying the adapted model, but also leak the private dataset to the LLM provider.

### 5.3. Transferring Prompts for Generation Tasks

Since, following Li et al. (2022), we model classification tasks as text-infilling tasks rather than adding a classification head, as detailed in Appendix B.2, our POST method naturally extends to generation tasks. To demonstrate this ability in practice we assess POST's performance on the MIT movie dataset where the task is to extract a movie's director or genre from a given movie description. Instead of generating a single token, as for classification, this open-ended task requires generating multiple tokens of varying lengths. We choose AIE, which aims to extract location from a note, as a public dataset and present our results in Table 2. POST outperforms Full ZS and Compressed PT, highlighting our method's applicability to open-ended tasks.

Table 1: **POST on Llama2-7b with and without DP.** We report POST's transfer performance. The baselines are the large models' zero-shot performance on the private data (Full ZS), the accuracy of tuning the prompt with the private data on the large models (Full PT) and the small model (Compressed PT), and the performance of the prompt tuned on the small model when direly applied to the large one (Direct Transfer). POST significantly improves performance over the small prompted model and our prompt transfer yields a strong improvement over the direct transfer. The same improvement can be observed for Roberta-base and GPT2-XL, as shown in Table 13 and Table 14 in the Appendix.

| Private | Full ZS | Full PT | Compressed PT | Direct Transfer | POST (ours) | | | |
| | | | | | Public | Test acc | Public | Test acc |
|---|---|---|---|---|---|---|---|---|
| sst2 | 78.67 | 94.84 | 78.78 | 55.28 | tweet | 89.33 | imdb | **90.02** |
| tweet | 44.50 | 72.03 | 54.12 | 41.70 | imdb | 57.55 | sst2 | **61.15** |
| arisetv | 76.57 | 93.47 | 77.92 | 54.23 | agnews | **86.71** | tweet | 79.59 |
| mpqa | 53.11 | 92.36 | 83.82 | 32.96 | sst2 | **87.37** | boolq | 76.18 |

(a) Confidential POST transfer (no DP).

| Private | Full ZS | Full PT | Compressed PT | Direct Transfer | POST (ours) | | | |
| | | | | | Public | Test acc | Public | Test acc |
|---|---|---|---|---|---|---|---|---|
| sst2 | 78.67 | 90.60 | 70.99 | 53.55 | tweet | 87.50 | imdb | **89.91** |
| tweet | 44.50 | 62.40 | 48.16 | 41.65 | imdb | 56.60 | sst2 | **59.55** |
| arisetv | 76.57 | 83.73 | 64.43 | 64.73 | agnews | **82.60** | tweet | 75.24 |
| mpqa | 53.11 | 85.95 | 68.87 | 33.28 | sst2 | 68.88 | boolq | **80.17** |

(b) Private POST transfer (with DP: $\varepsilon = 8$, and $\delta$ according to the dataset size as specified in Table 10).

Table 2: **POST on Generation Tasks (Llama2-7b, no DP).** We present POST's performance on the open-ended MIT-D and MIT-G tasks and choose the AIE dataset as the public set. We show its effectiveness over the baselines.

| Private | Full ZS | Full PT | Compressed PT | Direct Transfer | Public | Test Acc (POST) |
|---|---|---|---|---|---|---|
| MIT-D | 70.84 | 92.28 | 21.69 | 43.61 | AIE | **75.66** |
| MIT-G | 51.28 | 89.35 | 51.92 | 50.51 | AIE | **61.41** |

## 5.4. Practical Privacy Implications of POST

We further assess the practical privacy implications of POST. A standard method for analyzing privacy risks in machine learning is membership inference attacks (MIAs) (Shokri et al., 2017). In our setup, the goal of the attack is to determine whether a given data point was used to train the private soft prompt $p_s$. Such an attack could be executed by either the LLM owner or a third party to gain knowledge on $D_{pri}$. We instantiate the Likelihood Ratio Attack (LiRA) (Carlini et al., 2022) against soft prompts tuned on the four datasets for Roberta-base. As we show in Table 3, with as few as eight shadow models, LiRA achieves a non-trivial level of data leakage, with an AUC of 0.5610 for the sst2 dataset. Applying DP with $\varepsilon = 8$ and $\delta = 1.5 \times 10^{-5}$ significantly decreases the AUC to 0.5258. We also report TPR@1%FPR for the experiments. Results show that DP-tuned soft prompts reduce privacy leakage for all datasets. Beyond the privacy risk associated with the private soft prompt $p_s$, we also analyze the potential pretraining data leakage during the KD phase, as shown in Appendix D.7.

## 5.5. Runtime of POST

A notable advantage of POST is its ability to tune soft prompts on smaller models instead of significantly larger ones. Since tuning a soft prompt involves backpropagation

Table 3: **LiRA Membership Inference Attack.** We report the results of the LiRA attack against a soft prompt trained on four datasets with Roberta-base. Without DP ($\varepsilon = \infty$), we observe non-trivial leakage. Our method with DP reduces privacy leakage significantly in comparison to the non-private (confidential, $\varepsilon = \infty$) prompts.

| | sst2 | | imdb | | tweet | | arisetv | |
| | $\epsilon = \infty$ | $\epsilon = 8$ | $\epsilon = \infty$ | $\epsilon = 8$ | $\epsilon = \infty$ | $\epsilon = 8$ | $\epsilon = \infty$ | $\epsilon = 8$ |
|---|---|---|---|---|---|---|---|---|
| AUC | 0.5610 | 0.5258 | 0.5796 | 0.5524 | 0.5269 | 0.5006 | 0.5516 | 0.5491 |
| TPR@1% FPR | 0.0232 | 0.0137 | 0.0236 | 0.0157 | 0.0153 | 0.0128 | 0.0192 | 0.0181 |

through the entire model, this approach has the potential to substantially reduce computational complexity. In Table 4, we present a detailed runtime analysis of POST's individual components and compare them to full PT. Note that in practice, full PT on the large LLM leaks all the private prompt data to the LLM owner and is, hence, not practical. It serves only as a conceptual baseline.

Our findings reveal a significant speedup, particularly for large downstream datasets, where the cost of backpropagating many samples through a large language model is exceptionally high. For instance, on the sst2 dataset, our method achieves a sixfold speedup, reducing runtime from 2,660 minutes to just 409 minutes on an A100 GPU. These results highlight that POST not only enhances privacy but also delivers substantial computational efficiency improvements, especially for large datasets. Further details can be found in Appendix C.2.

## 5.6. Ablations

We perform various ablations on the inner workings of our method to provide guidance on how to best use it in practice.

Table 4: **Runtime of POST (GPT2-XL).** We present the runtime for our method, split by its individual components, and compare against full prompt tuning on the large LLM (PT on $\Phi_t$). We use arisetv and sst2 as private data and execute 5000 steps of transfer. We tune and transfer the prompts for 20 epochs (sst2) and 40 epochs (arisetv). All experiments are executed on a single A100 GPU.

| Method | Runtime for arisetv (min) | Runtime for sst2 (min) |
|---|---|---|
| PT on $\Phi_t$ | 368 | 2660 |
| (1) PT on $\Phi_s$ | 46 | 310 |
| (2) Transfer | 99 | 99 |
| **Ours total (1)+(2)** | **145** | **409** |
| (3) KD | 1203 | 1203 |
| **Ours total (1)+(2)+(3)** | **1348** | **1612** |

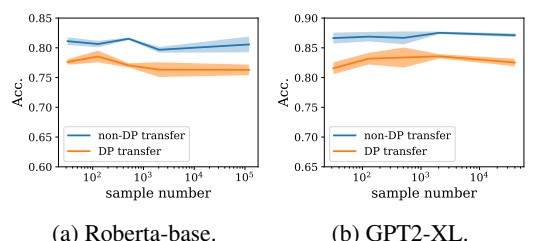

(a) Roberta-base.  (b) GPT2-XL.

Figure 2: **Effect of Number of Public Samples (arisetv).**

**Effect of Number of Public Samples used for Transfer.** We first investigate the influence of the size of the public dataset required to complete the prompt transfer. Our results in Figure 2 for arisetv as the private dataset with different numbers of public samples from agnews show that we can already yield high transfer performance with less than 100 public data points. The small size of the public datasets needed makes our method highly practical.

**Effect of Number of Transfer Steps.** We additionally investigate how many transfer steps are required to obtain good performance. Based on the insights from the previous section, we randomly subsample 128 samples from the agnews dataset as public data and report the achieved accuracy on arisetv as private data over different numbers of transfer steps. Our results in Figure 3 highlight that only a small number of transfer steps is enough for convergence and high accuracy on the private task. We observe convergence within around 2000 steps for GPT2-XL and around 1000 steps for Roberta-base.

**Choice of Public Dataset.** We observe that the choice in the public dataset impacts the final test performance. To evaluate this impact of the public dataset choice on transfer performance, we conduct an ablation study using different public sets for various private datasets, see Table 16. Our findings indicate that the best transfer performance can be achieved with public datasets from the same task family, even if the datasets are not very similar and have, for exam-

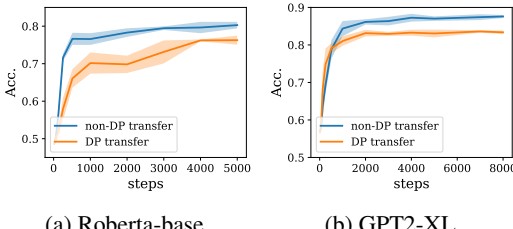

(a) Roberta-base.  (b) GPT2-XL.

Figure 3: **Effect of Number of Transfer Steps (arisetv).**

Table 5: **Baseline Comparison.** We compare POST against state-of-the-art baselines, DP-OPT (Hong et al., 2023)(**OPT, DP-OPT**) and Zero-Shot Transfer (Wu et al., 2023) (**ZST**). We report test accuracies over different private datasets. We use our distilled 2-layer compressed Llama2-7b (**2-Lay**) or **GPT2** as the source model, and Llama2-7b as the target model. For the DP versions, we use $\varepsilon = 8$ and $\delta$ according to Table 10. For our POST, we report the accuracies under the best public data (Table 13 and 14).

| Method | $\Phi_s$ | sst2 | imdb | tweet | arisetv | mpqa | disaster | trec |
|---|---|---|---|---|---|---|---|---|
| OPT | 2-Lay | 81.31 | 68.44 | 38.35 | 82.00 | 58.50 | 46.00 | **58.00** |
| OPT | GPT2 | 81.65 | 79.55 | 43.40 | 78.26 | 77.60 | 55.60 | 46.80 |
| DP-OPT | GPT2 | 72.59 | 69.53 | 24.90 | 30.44 | 61.80 | 48.90 | 32.60 |
| ZST | 2-Lay | 62.38 | 70.57 | 42.80 | 58.33 | 33.31 | 43.55 | 18.22 |
| ZST with DP | 2-Lay | 53.55 | 69.47 | 41.65 | 59,54 | 32.70 | 43.49 | 11.60 |
| **POST (ours)** | 2-Lay | **90.14** | **86.27** | **61.70** | **86.71** | **87.37** | **62.84** | 53.40 |
| **DP-POST (ours)** | 2-Lay | 89.91 | 83.26 | 59.55 | 82.60 | 80.17 | 58.62 | 37.80 |

ple, different numbers of classes. We added a more detailed discussion in Appendix D.1.

**Design of the KD.** Building on our intuition that more similar models exhibit better prompt transfer, we assess different ways of preserving this similarity during KD (see Equation (1)). We observe that fixing the language modeling head, *i.e.,* causing higher output similarity, leads to slightly better transfer performance. In contrast, we do not observe a consistent improvement with fixing the embeddings. Detailed results are shown in Appendix D.3.

**Impact of Model Compression.** We also investigate the trade-offs of different compression ratios. We distill the large LLM to various smaller sizes and report the performance of our POST and the distillation time. As we show in Appendix D.4, as the distilled model becomes larger, the transfer performance generally becomes better. However, it also requires more distillation time and more computational resources from the user to tune the soft prompt locally.

**Transfer Loss Design.** Our transfer loss in Equation (3) has a parameter $\alpha$ controlling the balance between the two loss terms. We observe that a good choice of $\alpha$ depends largely on the target model's ZS performance and the compressed model's performance. We provide a detailed evaluation and a heuristic for selecting the best $\alpha$ in Appendix D.5.

Table 6: **Impact of Feature Space Alignment on Transfer Performance.** We compare the transfer performance of a distilled 2-layer Llama model and a non-distilled model that is fine-tuned directly on the same BookCorpus dataset. Both models have the same architecture and initialization, but only the distilled model maintains feature alignment with the large model. While the two models perform similarly before prompt transfer, the distilled model consistently achieves higher prompt transfer performance.

| Dataset | PT on Distilled Model | PT on non-Distilled Model | Transfer with the Distilled Model | Transfer with the non-Distilled Model |
|---|---|---|---|---|
| sst2 | 78.78 | 78.24 | 90.02 | 85.38 |
| imdb | 79.95 | 79.40 | 87.29 | 82.45 |
| tweet | 54.12 | 54.79 | 61.15 | 45.65 |
| arisetv | 77.92 | 72.07 | 86.71 | 70.65 |
| mpqa | 83.82 | 84.16 | 87.37 | 81.52 |

**Effect of Soft Prompt Token Length.** We investigate the impact of token length on the transferability of the soft prompt by conducting experiments across various lengths ( Appendix D.6). Our findings suggest that an optimal range lies between 50 and 100 tokens. In contrast, both excessively short and long prompts lead to suboptimal results, with longer prompts also incurring higher computational costs.

**Impact of Feature Space Alignment.** In addition to the empirical results on prompt transfer, we further investigate why soft prompts trained on the distilled model remain transferable to the target model. Specifically, we conduct an ablation study to examine the role of feature space alignment induced by KD in our POST's transfer performance. We use a model with the same architecture and initial parameters as our distilled 2-layer Llama. Instead of applying knowledge distillation on the BookCorpus dataset, we directly fine-tune this model on BookCorpus to match the performance of the distilled model. This model has no feature space alignment with the large model. We then evaluate its transfer performance and compare it to that of the distilled model, following the same transferring process in Section 4.3. The results in Table 6 show that, although both models achieve similar prompt tuning performance, the prompt transfer performance for the distilled model is always better. This indicates that using knowledge distillation to preserve feature space alignment between the small model and large model is essential in our designed prompt transfer procedure.

### 5.7. Comparing against State-of-the-Art Prompt Transfer Approaches

We compare against two state-of-the-art baselines, namely the **Zero-Shot transfer** by Wu et al. (2023) and **DP-OPT** by Hong et al. (2023). Zero-Shot transfer operates in the same setup as we do and also relies on soft prompts. DP-OPT, in contrast to ours, is designed for discrete prompts. Details on their methods are presented in Appendix C.3. DP-OPT first tunes a discrete prompt locally and then directly uses it on the large model. Since their method relies on the small model having good performance, we execute their method in 2 setups for a fair comparison: 1) we tune their

source prompt using our compressed model as the small model, and 2) we use GPT2 as the small model. The latter is expected to have significantly higher performance and yield much better prompts. We report DP-OPT's performance on both non-DP (OPT) and DP settings. Our results in Table 5 highlight that our POST significantly outperforms all baselines even in the DP regime. Additional results where we distill from GPT2-XL can be found in Appendix C.3.

## 6. Conclusions

We present POST, a framework for the private transfer of soft prompts that enables adapting LLMs with private user data while protecting both the user's privacy and the LLM provider's intellectual property. POST relies on distillation to enable an LLM provider to share a small model with limited utility to a client for local prompt tuning on their private data, optionally with DP guarantees. Using our new prompt transfer method that leverages a small set of public data, the LLM provider can then transfer the prompt to their model. Our experiments highlight that the POST framework achieves significant improvements on the private tasks through the prompt transfer, improves the computational efficiency of prompt tuning and outperforms all private prompt transfer baselines. Thereby, our work paves the way for more trustworthy application of LLMs.

## Impact Statement

We propose a private transfer of soft prompts from a small language model to a large language model. The primary positive societal impact of our work is that our method can protect local data privacy and also the intellectual property of the large language model provider, which encourages wider and more trustworthy applications of LLMs. Additionally, since our transfer enables more compute-efficient prompt tuning and enables the reuse of existing prompts, it can have a positive environmental impact. We expect that our POST framework will also enable more LLM providers to offer the possibility of adapting their LLMs through soft prompt tuning.

## Acknowledgments

The project on which this paper is based was funded by the German Federal Ministry of Education and Research (BMBF) under funding number 16KIS2114K. Responsibility for the content of this publication lies with the authors.

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

# A. Limitations

Our work proposes a method to protect the privacy and confidentiality of private data during the prompt tuning phase. However, we didn't address the privacy leakage risk during the inference phase. Also, this work does not aim to address the challenge of preventing pretraining data leakage. Although our experiments suggest that the distillation process results in minimal leakage, a more thorough investigation is warranted. Also, compression of the LLMs through knowledge distillation techniques may be computationally expensive for LLM providers. Additionally, in our method, the selection of a public dataset will affect the transfer performance of soft prompts. While we observe, in general, that public datasets that have a similar structure to the private data work best for transfer, there is no ideal strategy for selecting the optimal public dataset.

# B. Experimental Setup

## B.1. Knowledge Distillation

We follow the procedure of (Sanh et al., 2019) to initialize and distill our compressed model. We use the first and last layers of Roberta-base, the first two and last two layers of GPT2-XL, and the first and last layers of Llama2-7b to initialize our compressed Roberta-base, GPT2-XL and Llama2-7b before knowledge distillation. We also initialize the small student model's word embedding and language modeling head the same as their teacher model. We conduct experiments on whether to freeze the language modeling head and/or word embedding during knowledge distillation in Appendix D.3. Each model's structure and size are listed in Table 7.

Table 7: **Model Size before and after Distillation.**

| Model | Layer Number | Hidden Dimension | Head Number | Parameter Num (M) |
|---|---|---|---|---|
| Roberta-base | 12 | 768 | 12 | 125 |
| Our distilled Roberta-base | 2 | 768 | 12 | 53 |
| GPT2-XL | 48 | 1600 | 25 | 1560 |
| Our distilled GPT2-XL | 4 | 1600 | 25 | 205 |
| Llama2-7b | 32 | 4096 | 32 | 6738 |
| Our distilled Llama2-b | 2 | 4096 | 32 | 667 |

During knowledge distillation, we use the BookCorpus (Zhu et al., 2015) dataset, and we take the checkpoint model that is distilled for 50,0000 steps. The hyperparameters used in knowledge distillation are shown in Table 8.

Table 8: **Hyperparameters in Knowledge Distillation.**

| $\alpha_{ce}$ | $\alpha_{lm}$ | $\alpha_{cos}$ | Learning Rate | Batch Size |
|---|---|---|---|---|
| 5.0 | 2.0 | 1.0 | 0.00025 | 5 |

## B.2. Datasets and Tasks

We use the following datasets as private sets or transfer sets in our experiments:

- **sst2**: The Stanford Sentiment Treebank (sst2) is a binary sentiment classification dataset containing sentences from movie reviews, labeled as positive or negative.

- **imdb**: A large-scale sentiment classification dataset consisting of movie reviews from IMDb, labeled as positive or negative.

- **tweeteval**: A benchmark dataset for sentiment analysis on tweets, providing real-world social media text challenges. In our experiments, we use only the sentiment classification subset which contains 3 classes: negative, positive, and neutral.

- **arisetv**: A topic classification dataset derived from TV news transcripts, categorizing documents into 6 topic labels: business, sports, politics, health, entertainment, and tech.

- **agnews**: A widely used topic classification dataset containing news articles labeled into four categories: World, Sports, Business, and Science/Technology.

- **mpqa**: The Multi-Perspective Question Answering (MPQA) dataset, designed for sentiment and subjectivity analysis with fine-grained opinion annotations.

- **MIT**: A dataset for extracting attributes such as movie directors and genres from movie descriptions.

- **AIE**: A dataset for extracting locations from notes.

- **trec**: The TREC Question Classification Dataset is a benchmark dataset designed for question classification tasks in natural language processing (NLP). It is derived from the TREC (Text REtrieval Conference) Question Answering (QA) track, where the goal is to classify questions into predefined categories based on their topic.

- **disaster**: A dataset containing tweets related to natural disasters, used for binary classification of whether a tweet is disaster-related or not.

- **boolq**: The BoolQ (Boolean Questions) dataset is a benchmark dataset for yes/no question answering (QA). BoolQ is widely used for training and evaluating machine reading comprehension (MRC) models and is part of the SuperGLUE benchmark.

We follow Li et al. (2022) to formulate our classification tasks as a **text-infilling task** (instead of adding a classification head to output class probabilities). This means that for masked language models such as Roberta, we append "it was ¡mask¿" to the input and let the model predict the ground truth text. The setting for GPTs and Llama2 is similar in that we append "it was" to predict the next word. To increase the robustness of this method, we use multiple ground truth text labels and compare the average probability of outputting those text labels. See Table 9 for task templates and the ground truth labels used in our experiment.

Table 9: **Task Template and Ground Truth Labels used in Text-infilling.** means the sentence used in the dataset.

| Dataset | Task Template Roberta | Task Template GPT2/Llama2 | Ground Truth Text Label |
|---|---|---|---|
| sst2 | , it was <mask> | , it was | 0: [" terrible"," negative"," bad"," poor"," awful"]
1: [" positive"," good"," great"," awesome"," brilliant"," amazing"] |
| imdb | , it was <mask> | , it was | 0: [" terrible"," negative"," bad"," poor"," awful"]
1: [" positive"," good"," great"," awesome"," brilliant"," amazing"] |
| tweet | , it was <mask> | , it was | 0: [" terrible"," negative"," bad"," poor"," awful"]
1: [" moderate"," neutral"," balanced"]]
2: [" positive"," good"," great"," awesome"," brilliant"," amazing"] |
| arisetv | , it was about <mask> | , it was about | 0: [" business"], 1: [" sports"], 2: [" politics"]
3: [" health"],4: [" entertainment"],5: [" technology"," science"] |
| mpqa | , it was <mask> | , it was | 0: [" terrible"," negative"," bad"," poor"," awful"]
1: [" positive"," good"," great"," awesome"," brilliant"," amazing"] |
| disaster | , is this sentence related to disaster? <mask> | , is this sentence related to disaster? | 0:[" no"," No"],1:[" yes", "Yes"] |
| trec | , this question is about <mask> | , the topic of the question is | 0:[" abbreviation"],1:[" entity"],2: [" description"], 3:[" human"],
4:[" location"],5:[" numerical value"] |

## B.3. Prompt Tuning

Following Su et al. (2022)'s setting, we use the soft prompt with a length of 100 tokens in all our experiments. We follow Duan et al. (2023a)'s setting to obtain DP private prompts with PromptDPSGD. Table 10 shows the hyperparameters used in this experiment.

Table 10: **Hyperparameters used for Prompt Tuning, including the $\delta$ for PromptDPSGD.**

| Dataset | $\delta$ | Epochs | Learning Rate |
|---|---|---|---|
| sst2 | $1.5 \times 10^{-5}$ | 20 | 0.001 |
| imdb | $4 \times 10^{-5}$ | 20 | 0.001 |
| tweet | $2 \times 10^{-5}$ | 20 | 0.001 |
| arisetv | $2 \times 10^{-4}$ | 40 | 0.001 |
| mpqa | $1.16 \times 10^{-4}$ | 20 | 0.001 |

## B.4. Public Datasets for Prompt Transfer

We rely on small public datasets to perform our prompt transfer. A question is the right choice of the public dataset. We normally choose the public dataset that performs a similar task as the private dataset, such as choosing imdb or tweet as the public dataset of sst2 as they are all sentiment classification tasks. Transferring with a public dataset that performs a different task from the private dataset may lead to suboptimal performance. We tested this setting to transfer soft prompts trained on arisetv, a topic prediction dataset. The transfer performance of using tweet as public dataset is acceptable but generally worse than using agnews, another topic prediction dataset, as a public dataset. In general, we found that the public and private datasets do not need to have the same structure, such as the class number. For example, using tweet (3 classes) as a public dataset leads to better transfer performance than imdb (2 classes) on sst2 (also 2 classes). This highlights the robustness of our method and the broad selection of public datasets for the transfer.

We report the hyperparameters used in the transfer experiments as Tables 11 and 12.

Table 11: **Hyperparameters used during Prompt Transfer.**

| Model | Batch Size | Optimizer | Learning Rate |
|---|---|---|---|
| Roberta-base | 32 | Adam | 0.001 |
| GPT2-XL | 8 | Adam | 0.001 |
| Llama2-7b | 4 | Adam | 0.0005 |

Table 12: **Setting of $\alpha$ for Different Datasets and Models during Prompt Trasnfer.**

| Model | Dataset | | | |
|---|---|---|---|---|
| | sst2 | imdb | tweet | arisetv |
| Roberta-base | 0.8 | 0.8 | 0.2 | 0.4 |
| GPT2-XL | 0.7 | 0.4 | 0.0 | 0.6 |
| Llama2-7b | 0.9 | 1.0 | 0.6 | 0.6 |

# C. Additional Experiments

## C.1. Models of Different Scales

We present a full overview of our POST method's results on LLMs with different scales, *i.e.,* Roberta-base and GPT2-XL, in Table 13 and Table 14 for confidential and DP transfer, respectively. We observe that over all models with different scales, our method significantly outperforms the baselines, both without and with DP.

## C.2. Runtime

In Table 4, we analyze the runtime of our POST's individual components and compare them to the runtime of full prompt tuning on GPT2-XL. Note that in practice, full prompt tuning on the large LLM leaks all the private prompt data to the LLM owner and is, hence, not practical.

Prompt tuning time, for both large and small models, depends on the size of the private dataset when backpropagating over all private training examples. To illustrate this, we report results for both a small dataset ( with 4.7k training samples) and a large dataset (sst2 with 67.3k training samples). KD is a one-time operation performed by the LLM owner, after which the distilled model can be used by multiple users. As a result, the compute cost of KD amortizes over time. To account for this, we present two sets of results for our method: (1) only the prompt tuning on the small model and the transfer step, and (2) including the non-amortized KD time as a worst-case scenario. The runtime results in Table 4 show that, when considering only steps (1) and (2), our method significantly reduces runtime compared to full prompt tuning for both small and large datasets. Notably, for the larger sst2 dataset, our method achieves a 6x speedup, reducing runtime from 2660 minutes to just 409 minutes on an A100 GPU. In the worst-case scenario, where the non-amortized KD time is included and only one task is tuned (a highly unrealistic assumption for an LLM owner serving multiple users), prompt tuning on the small model is faster. Yet, for the large dataset, our method still outperforms full prompt tuning with a runtime of 1612 minutes compared to 2660 minutes. These results demonstrate that beyond enhancing privacy, our POST method also provides substantial improvements in computational efficiency, especially for large datasets.

Table 13: **Confidential Prompt Transfer Performance.** We compress Roberta-base and GPT2-XL, tune prompts for different private datasets on the compressed models, and transfer them back using different public datasets (POST). As baselines, we present the large models' zero-shot performance on the private data (Full ZS), the accuracy of tuning the prompt with the private data on the large models (Full PT) and the small model (Compressed PT), and the performance of the prompt tuned on the small model when direcly applied to the large one (Direct Transfer). Our POST significantly improves performance over the small prompted model, and our prompt transfer yields a strong improvement over the direct transfer.

| Private | Full ZS | Full PT | Compressed PT | Direct Transfer | POST (ours) | | | |
|---------|---------|---------|---------------|-----------------|--------|----------|--------|----------|
| | | | | | Public | Test acc | Public | Test acc |
| sst2 | 72.25 | 91.74 | 79.10 | 76.49 | tweet | **87.16** | imdb | 85.21 |
| imdb | 72.19 | 89.88 | 78.85 | 76.92 | tweet | **83.65** | sst2 | 80.27 |
| tweet | 36.53 | 68.68 | 56.65 | 43.10 | imdb | 54.55 | sst2 | **59.65** |
| arisetv | 38.80 | 89.81 | 70.98 | 47.82 | agnews | **82.97** | tweet | 68.48 |

(a) Roberta-base.

| Private | Full ZS | Full PT | Compressed PT | Direct Transfer | POST (ours) | | | |
|---------|---------|---------|---------------|-----------------|--------|----------|--------|----------|
| | | | | | Public | Test acc | Public | Test acc |
| sst2 | 60.78 | 94.84 | 80.94 | 59.06 | tweet | **85.89** | imdb | 83.49 |
| imdb | 60.27 | 93.28 | 81.32 | 60.34 | tweet | **83.65** | sst2 | 82.15 |
| tweet | 34.71 | 68.60 | **63.13** | 41.50 | imdb | 60.55 | sst2 | 57.70 |
| arisetv | 52.98 | 92.45 | 77.10 | 55.43 | agnews | **87.56** | tweet | 82.12 |

(b) GPT2-XL.

## C.3. Baseline Comparison

Zero-Shot transfer performs prompt transfer by using the embeddings of some tokens from the vocabulary as a form of support vector to transform the source prompts into a relative space, and then search for the corresponding target prompt embeddings for the target model. To provide the optimal source model for their approach, we use a compressed model that we obtained by keeping the embedding layer frozen during KD (see row 3 in Table 18). DP-OPT, in contrast to ours, is designed for discrete prompts. They first tune a discrete prompt locally and then directly use it on the large model.

We also run the baseline comparison on the GPT2-XL model, as presented in Table 15. Our method consistently outperforms other methods on GPT2-XL.

## D. Additional Ablations

### D.1. Choice of Public Dataset

We investigate the effect of the choice of public data in Table 16. The datasets fall into three categories: sentiment classification (sst2, imdb, tweet, mpqa), disaster detection (disaster), and topic classification (arisetv, agnews). The results show that using the same dataset for both private and public sets consistently achieves the best performance. Moreover, datasets with similar tasks transfer well, even if their number of classes differs. For example, imdb (2 classes) achieves 83.65 using tweet (3 classes) as the public set, and arisetv (6 classes) performs well with agnews (4 classes), *i.e.,* 87.56 accuracy, indicating that task similarity is more crucial than class alignment. In contrast, using a public dataset from a different domain degrades performance. Sentiment classification datasets perform poorly when paired with disaster which aims to detect disaster, as seen in tweet (48.15). Likewise, sst2 and imdb show weak results when using arisetv or agnews. These findings highlight that selecting a public dataset with a similar task to the private task is helpful for effective prompt transfer.

### D.2. Other Model Compression Techniques

In this work, we focus on KD to compress the LLM. In the following, we discuss why alternative compression techniques are less suited to implement our POST framework.

**Quantization** (Han et al., 2016; Jacob et al., 2018) reduces the precision of model parameters to lower bit-widths, effectively decreasing model size. However, since our framework involves transmitting the proxy model to the user, quantization primarily reduces storage requirements without adequately protecting the intellectual property inherent in the original large model. Additionally, quantized models often require specialized hardware or software support to maintain performance, which may not be available to all users.

**Pruning** (Han et al., 2015; Li et al., 2017; Wang et al., 2024) involves removing less significant weights or neurons from the

Table 14: **Differentially Private and Confidential Prompt Transfer Performance.** We compress Roberta-base and GPT2-XL, tune prompts for different private datasets on the compressed models with Differential Privacy guarantees ($\varepsilon = 8$), and transfer them back using different public datasets (POST). As baselines, we present the large models' zero-shot performance on the private data (Full ZS), the accuracy of PromptDPSGD tuned prompt with the private data on the large models (Full PT) and the small model (Compressed PT), and the performance of the prompt tuned on the small model when direcly applied to the large one (Direct Transfer). Our POST significantly improves performance over the small prompted model and our prompt transfer yields a strong improvement over the direct transfer.

|         |         |         |               |                 |        | POST (ours) |        |          |
|---------|---------|---------|---------------|-----------------|--------|-------------|--------|----------|
| Private | Full ZS | Full PT | Compressed PT | Direct Transfer | Public | Test acc    | Public | Test acc |
| sst2    | 72.25   | 90.14   | 67.54         | 77.06           | tweet  | **84.40**   | imdb   | 81.42    |
| imdb    | 72.19   | 88.55   | 72.22         | 74.35           | tweet  | 79.64       | sst2   | **80.64**|
| tweet   | 36.53   | 62.05   | 40.87         | 43.15           | imdb   | 55.65       | sst2   | **59.25**|
| arisetv | 38.80   | 80.33   | 64.25         | 47.34           | agnews | **79.11**   | tweet  | 71.98    |

(a) Roberta-base.

|         |         |         |               |                 |        | POST (ours) |        |          |
|---------|---------|---------|---------------|-----------------|--------|-------------|--------|----------|
| Private | Full ZS | Full PT | Compressed PT | Direct Transfer | Public | Test acc    | Public | Test acc |
| sst2    | 60.78   | 91.28   | 74.31         | 57.80           | tweet  | 79.93       | imdb   | **84.06**|
| imdb    | 60.27   | 89.59   | 74.81         | 63.66           | tweet  | **78.03**   | sst2   | 75.16    |
| tweet   | 34.71   | 61.47   | 48.60         | 41.50           | imdb   | **58.05**   | sst2   | 54.75    |
| arisetv | 52.98   | 83.24   | 67.16         | 57.25           | agnews | **82.12**   | tweet  | 80.55    |

(b) GPT2-XL.

Table 15: **Baseline Comparison.** We present the performance of our method against state-of-the-art baselines on GPT2-XL. We use **our compressed** to represent the 4-layers small model distilled from GPT2-XL. Our method outperforms both baselines.

| Method | $\Phi_t$ | $\Phi_s$ | sst2 | imdb | tweet | arisetv |
|--------|----------|----------|------|------|-------|---------|
| OPT (Hong et al., 2023) | GPT2-XL | our compressed | 60.67 | 61.70 | 30.70 | 42.87 |
| OPT (Hong et al., 2023) | GPT2-XL | GPT2 | 62.16 | 63.18 | 35.20 | 46.38 |
| Zero-Shot Transfer (Wu et al., 2023) | GPT2-XL | our compressed | 63.65 | 61.27 | 41.60 | 56.64 |
| Zero-Shot Transfer (Wu et al., 2023) with DP | GPT2-XL | our compressed | 63.42 | 61.71 | 41.35 | 57.25 |
| **POST (ours)** | GPT2-XL | our compressed | **85.89** | **83.93** | **61.75** | **87.56** |
| **DP-POST (ours)** | GPT2-XL | our compressed | 84.06 | 78.03 | 58.05 | 82.12 |

model. Unstructured pruning, which eliminates individual weights, typically does not lead to practical speedups or reduced resource consumption, making it less applicable to our approach. Structured pruning, on the other hand, removes entire neurons, filters, or layers, resulting in a smaller and more efficient model. However, many structured pruning methods are task-specific and require fine-tuning on specific tasks to regain performance.

In addition to the disadvantages of the other methods that make them less suited for our POST framework, KD also yields a desirable property for our purpose, namely, it aligns the outputs between the large LLM and the small model. This alignment in output also positively impacts the alignment of internal model behavior, supporting a better transferability of prompts tuned on the small model to the large LLM, as presented in Table 6.

### D.3. Knowledge Distillation Design

**Knowledge Distillation Loss.** The knowledge distillation loss, as shown in Equation (1), consists of three terms. We conducted an ablation study to analyze the impact of including or excluding each loss term and compared their end-to-end transfer performance using the Roberta-base model, as presented in Table 17. The results demonstrate that combining all three loss terms consistently yields the best transfer performance, validating our choice of loss function for KD.

**Knowledge Distillation Setup.** We also investigated the best way of performing KD to improve prompt transferability. In particular, we analyzed the impact of keeping the word embedding or(and) language modeling heads frozen during KD on the prompt transfer performance. Our results in Table 18 highlight that keeping the language modeling head fixed performs slightly better than the alternative which mainly performs on-par. These results indicate that the successful transfer of our method is robust to the KD and independent of any specific KD setting.

**When to Stop Distilling.** Since the LLM provider does not have any knowledge of the users' private downstream tasks, they

Table 16: **Ablation on the Choice of Public Set.** We observe that best transfer performance can be observed when using same-domain data. In particular, it helps to have data from the same *task familiy*.

| Private Set | Public Set | | | | | | |
|---|---|---|---|---|---|---|---|
| | sst2 | imdb | tweet | mpqa | disaster | arisetv | agnews |
| sst2 | **87.27** | 85.21 | 87.16 | 82.68 | 85.89 | 76.15 | 76.72 |
| imdb | 80.27 | **86.67** | 83.65 | 76.50 | 78.99 | 76.65 | 76.42 |
| tweet | 59.65 | 54.55 | **63.55** | 51.15 | 48.15 | 54.00 | 59.50 |
| arisetv | 82.12 | 77.41 | 78.62 | 50.84 | 57.13 | 86.47 | **87.56** |

Table 17: **Ablation on Knowledge Distillation Loss.** We compared the end-to-end prompt transfer performance of having each loss term and their combination in the knowledge distillation. We first distill the model to the same checkpoint with different loss combinations, and then compare their transfer performance. We find that our choice of 3-way loss achieves best performance.

| | $\mathcal{L}_{ce}$ | $\mathcal{L}_{lm}$ | $\mathcal{L}_{cos}$ | $\mathcal{L}_{ce} + \mathcal{L}_{lm}$ | $\mathcal{L}_{ce} + \mathcal{L}_{cos}$ | $\mathcal{L}_{lm} + \mathcal{L}_{cos}$ | $\mathcal{L}_{ce} + \mathcal{L}_{lm} + \mathcal{L}_{cos}$ |
|---|---|---|---|---|---|---|---|
| sst2 | 85.32 | 78.21 | 76.38 | 85.67 | 84.75 | 85.67 | **87.73** |
| imdb | 83.17 | 79.05 | 75.02 | 81.26 | 82.26 | 80.12 | **83.96** |
| tweet | 53.65 | 47.05 | 45.80 | 54.95 | 55.05 | 50.40 | **58.25** |
| arisetv | 78.99 | 67.87 | 75.60 | 74.52 | 82.49 | 70.65 | **82.73** |

Table 18: **Analyzing the KD Setup.** We perform an ablation on different designs of the KD and present their impact on the prompt transfer for the private arisetv dataset, using agnews as public data. We analze different combinations of freezing the embedding (Fix emb) and freezing the language modeling head (Fix head).

| Model | Fix emb | Fix head | Acc. | Model | Fix emb | Fix head | Acc. |
|---|---|---|---|---|---|---|---|
| roberta-base | ✗ | ✓ | 81.68 ±0.764 | GPT2-XL | ✗ | ✓ | 87.52 ±0.505 |
| | ✓ | ✓ | 80.79 ±0.885 | | ✓ | ✓ | 86.51 ±0.726 |
| | ✓ | ✗ | 80.84 ±0.360 | | ✓ | ✗ | 86.81 ±0.732 |
| | ✗ | ✗ | 80.11 ±0.738 | | ✗ | ✗ | 87.48 ±0.170 |

cannot rely on any external signal as a termination criterion for their KD. Given that they can observe the distillation loss, we find that terminating the KD when the loss starts plateauing is easy to implement and effective. Hence, as the small model, we select the checkpoints of 500,000 steps, which are the checkpoints directly after the loss plateaus. In Figure 4, we analyze the relationship between distillation loss and steps. We also conduct transfer experiments with different checkpoints and show the relation between distillation loss and transfer performance in Figure 5b. We observe that the performance gain becomes smaller as the distilled model starts to converge. Therefore, by finishing the KD as early as possible, we save computing time while still being effective.

## D.4. Influence of Compressed Model Size

In Table 19, we also compare the transfer performance from distilled models with different compression ratios (in our case, the number of layers). These results highlight that as the distilled model becomes larger, the transfer performance generally becomes better. However, it also requires more distillation time and more computational resources from the user to tune the soft prompt locally. We observe that the gain that can be achieved through soft prompt transfer becomes smaller or even negative for datasets like imdb and tweet, with a 3-layer distilled model. This would incentivize the users to not transfer their prompts back to the large model, thereby disrupting the LLM provider's business model. Hence, we found that our choice of a 2-layer (4-layer) compressed model for Roberta-base (GPT2-XL) offers a good balance for the LLM provider and user.

To study the relationship between transfer performance (evaluated by downstream task accuracy) and performance of the compressed model (evaluated by checkpoint loss) further, we also conducted ablations where we compress the models to different numbers of layers and with different distillation steps. We present the results in Figure 5a and Figure 5b. They highlight that overall better compressed models lead to better transfer accuracy.

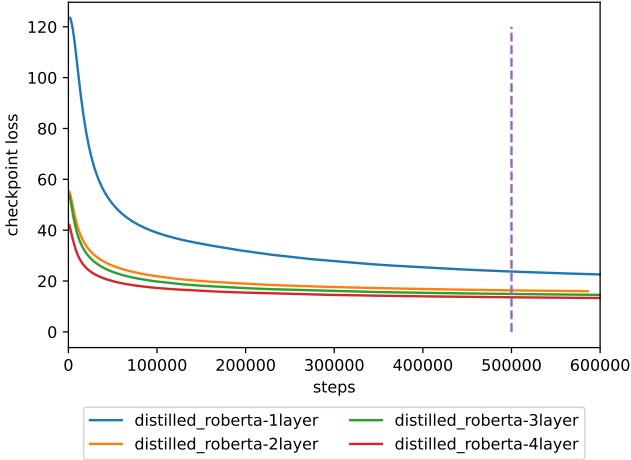

Figure 4: **Checkpoint Loss on Training Steps for Distilled RoBERTa Models with Varying Numbers of Layers.** The plot illustrates the convergence behavior of 1-layer, 2-layer, 3-layer, and 4-layer distilled models. The dashed vertical line represents the point at which a specific checkpoint is selected for evaluation in our experiments.

Table 19: **Influence of Compressed Model Size (Number of Layers).** We compare the confidential prompt transfer performance of compressed models based on RoBERTa-base, varying the number of layers in the compressed model. We also report the compressed model performance in parentheses. As the number of layers increases, transfer performance generally improves, but the distillation time also increases. Our selected number of layers in the compressed models offers a good balance between transfer performance and distillation cost.

| # layers in distilled version | 1 layer | 2 layers (paper) | 3 layers |
|---|---|---|---|
| sst2 | 84.52 (71.67) | 87.73 (78.78) | 88.53 (85.55) |
| imdb | 78.01 (71.03) | 83.96 (79.95) | 83.64 (83.88) |
| tweet | 50.65 (41.84) | 54.55 (54.12) | 61.50 (66.66) |
| arisetv | 53.62 (24.55) | 82.73 (77.92) | 86.45 (84.77) |
| Distill time | 6h 04min | 6h 45min | 7h 35min |

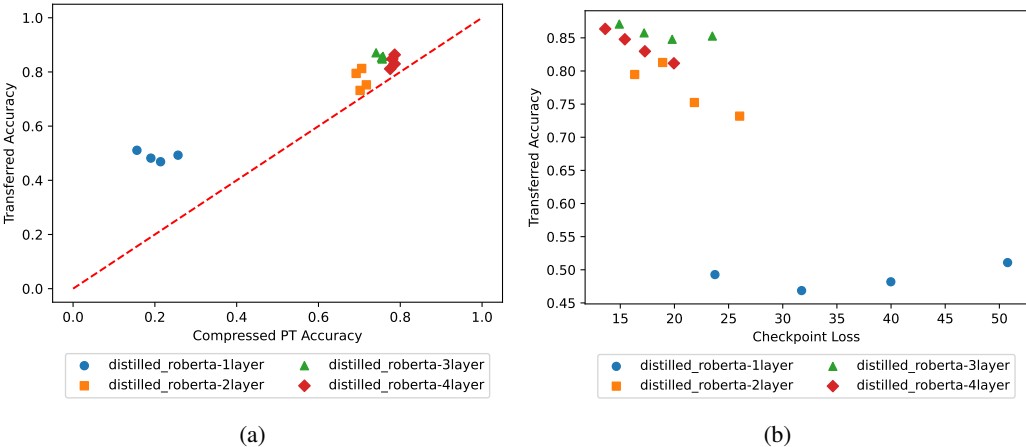

(a)                                                    (b)

Figure 5: **Analysis of Transferred Accuracy versus Compressed PT Accuracy and Checkpoint Loss at 50,000, 100,000, 200,000 and 500,000 Distillation Steps.** (a) compares transferred accuracy to compressed PT accuracy for different distilled Roberta models. (b) compares transferred accuracy to checkpoint loss for different distilled Roberta models.

### D.5. Influence of Transfer Loss Design

We further conduct an ablation study to examine the impact of the choice of $\alpha$ during the prompt transfer. As described in Section 4.3, we should use a larger $\alpha$ when the teacher LLM $\Phi_t$ already has a non-trivial zero-shot performance on the private task compared to the compressed model. And we use smaller $\alpha$ when the compressed model has better performance. The intuition is that if the target model performs well, it needs to less mimic the behavior of the smaller model, but only incorporate that model's direction change induced by the prompt. On the other side, when $\Phi_t$ has poor zero-shot performance, we put more emphasis on the output of the compressed model to provide the update.

Based on this insight, we design a heuristic to find an optimal $\alpha$ as shown in Equation (6).

$$\tilde{\alpha} = min(\frac{ZS - RG}{C - RG}, 1.0), \tag{6}$$

where $ZS$ represents the zero-shot performance, $C$ represents the performance of the compressed model, and $RG$ represents the random guess performance of the dataset (*e.g.,* for sst2 with 2 classes, $RG$ is 0.5). This heuristic can serve as a guideline for choosing the best alpha for transfer. In Table 20, we perform a wide range of experiments using different alphas and compare the best-performing alpha with the one output by our heuristic. We observe that the heuristic successfully identifies a good range for $\alpha$ and is usually not far off the empirical optimal values. Finally, this wide hyperparameter search shows that multiple $\alpha$ values achieve good transfer performance, highlighting our method's robustness to the choice of $\alpha$.

Table 20: **Performance of Roberta-base, GPT2-XL and Llama2-7B on 4 Different Datasets with Various $\alpha$ Values.** Transferred accuracy are presented as mean (standard deviation). We observe that multiple $\alpha$ values achieve good transfer performance, highlighting the robustness of our POST to the choice of $\alpha$.

| $\alpha$ | Roberta-base | | | | GPT2-XL | | | | Llama2-7b | | | |
|---|---|---|---|---|---|---|---|---|---|---|---|---|
| | sst2 | imdb | tweet | arisetv | sst2 | imdb | tweet | arisetv | sst2 | imdb | tweet | arisetv |
| heuristic | 0.76 | 0.77 | 0.14 | 0.41 | 0.35 | 0.33 | 0.05 | 0.60 | 1.00 | 1.00 | 0.54 | 0.97 |
| 0.0 | 80.54 (1.23) | 78.22 (0.55) | 57.75 (1.88) | 83.41 (0.91) | 75.92 (0.53) | 70.67 (5.18) | **58.70 (1.61)** | 84.50 (2.29) | 69.57 (2.64) | 75.37 (0.44) | 60.08 (0.65) | 81.00 (0.37) |
| 0.1 | 81.94 (1.87) | 79.20 (2.92) | 57.68 (1.95) | 83.09 (1.45) | 76.68 (0.46) | 70.22 (3.45) | 58.48 (1.31) | 85.67 (0.25) | 68.43 (5.40) | 76.27 (2.07) | 59.58 (2.42) | 81.00 (0.37) |
| 0.2 | 83.33 (0.78) | 80.89 (1.85) | **58.13 (2.16)** | 83.62 (0.77) | 76.80 (0.65) | 78.39 (2.37) | 57.53 (0.88) | 85.23 (0.61) | 70.76 (4.70) | 76.12 (1.68) | 61.33 (0.99) | 79.83 (2.06) |
| 0.3 | 84.67 (0.73) | 80.57 (1.16) | 58.08 (1.15) | 83.62 (1.26) | 78.29 (1.75) | 56.98 (2.41) | 86.11 (1.15) | 73.78 (2.81) | 75.62 (0.70) | **61.62 (0.38)** | 81.32 (3.56) |
| 0.4 | 86.81 (0.46) | 79.86 (1.20) | 54.77 (1.04) | **83.94 (0.79)** | 76.53 (0.26) | 81.57 (1.88) | 51.42 (1.63) | 86.35 (0.36) | 70.18 (4.83) | 75.29 (0.04) | 59.77 (1.28) | 82.81 (0.87) |
| 0.5 | 88.61 (0.46) | 82.16 (1.55) | 54.10 (0.74) | 80.11 (1.73) | 76.83 (0.65) | **81.74 (0.71)** | 57.50 (2.09) | 87.24 (1.50) | 85.13 (4.34) | 80.24 (2.19) | 60.23 (1.34) | 83.45 (0.53) |
| 0.6 | **88.61 (0.78)** | 81.82 (1.06) | 52.88 (0.44) | 76.57 (0.84) | 79.82 (1.86) | 53.93 (0.64) | 87.72 (0.39) | 89.68 (0.20) | 81.90 (3.10) | 55.13 (0.18) | **85.47 (1.61)** |
| 0.7 | 87.88 (0.29) | 80.89 (1.59) | 52.33 (0.20) | 71.26 (0.64) | 85.86 (1.56) | 78.18 (0.93) | 54.27 (1.05) | 87.52 (0.28) | 89.18 (0.70) | 82.14 (1.72) | 52.12 (0.60) | 84.30 (4.75) |
| 0.8 | 87.35 (0.24) | **82.65 (0.95)** | 51.32 (0.24) | 68.68 (0.57) | **86.66 (0.93)** | 77.23 (2.07) | 50.53 (2.49) | 86.11 (1.62) | 89.22 (0.53) | 83.86 (0.55) | 52.25 (0.52) | 83.98 (1.76) |
| 0.9 | 86.47 (0.64) | 81.52 (1.39) | 49.38 (0.55) | 63.04 (0.85) | 84.33 (1.26) | 76.13 (0.91) | 48.52 (1.53) | 83.66 (1.42) | 88.69 (0.69) | 85.00 (2.35) | 49.50 (0.31) | 83.33 (2.06) |
| 1.0 | 85.82 (0.85) | 80.57 (1.32) | 48.42 (1.13) | 58.57 (0.24) | 83.37 (0.92) | 73.62 (1.33) | 48.42 (0.78) | 80.60 (1.35) | 86.47 (1.62) | **85.85 (1.77)** | 47.57 (1.13) | 84.02 (0.91) |

### D.6. Influence of Token Number of Soft Prompt

We follow (Wang et al., 2022) to use 100 tokens in our experiment. We conduct further experiments to investigate how token length influences the transferability of the soft prompt. We compare the transfer performance of soft prompts with lengths of 5, 10, 20, 50, 75, 100, and 200 as shown in Table 21. We find that a prompt length of 50-100 tokens is a good range, and our method is not very sensitive to the token length. Both too few or too many tokens in the soft prompt lead to sub-optimal performance. Additionally, too many tokens need more computing resources.

Table 21: **Ablation on the Token Number of Soft Prompt.** We compare the prompt transfer performance of different soft prompt token lengths on four classification tasks with Roberta-base.

| Token Num | 5 | 10 | 20 | 50 | 75 | 100 | 200 |
|---|---|---|---|---|---|---|---|
| sst2 | 84.17 | 85.66 | 86.35 | 87.38 | 86.70 | 87.16 | 85.32 |
| imdb | 75.95 | 76.53 | 80.42 | 82.77 | 83.27 | 83.65 | 79.92 |
| tweet | 55.25 | 57.60 | 57.80 | 57.00 | 60.00 | 59.65 | 60.40 |
| arisetv | 79.02 | 79.63 | 80.47 | 82.04 | 82.77 | 82.97 | 83.13 |

### D.7. Pretraining Data Leakage during Distillation

We also consider and investigate the pertaining data leakage during the KD phase. We follow the setup of Shi et al. (2023) to assess pretraining data leakage during the knowledge distillation in our paper, using the WikiMIA dataset. Specifically, we use the Pythia 2.8B model (Biderman et al., 2023) as our base model, as its pretraining data is publicly available. We

apply knowledge distillation to obtain two smaller student models: one with 3 layers and another with 4 layers. To evaluate data leakage, we conduct membership inference attacks (MIA) using the Mink% method and report the AUC-ROC and TPR@1%FPR for all three models. The results, summarized in Table 22, indicate that our knowledge distillation reduces the pretraining data leakage compared to the original model.

Table 22: **Ablation on Pretraining Data Leakage.** We compare AUC-ROC and TPR@1%FPR using the original Pythia 2.8B model and two distilled models on the WikiMIA dataset. The distilled models exhibit less pretraining data leakage compared to the original pretrained model.

|            | Pythia 2.8B | 3-layer Distilled Model | 4-layer Distilled Model |
|------------|-------------|-------------------------|-------------------------|
| AUC-ROC    | 0.7103      | 0.5737                  | 0.5750                  |
| TPR@1%FPR  | 0.0980      | 0.0196                  | 0.0196                  |

