# OpenReview forum: "Efficient and Privacy-Preserving Soft Prompt Transfer for LLMs"
_ICML.cc/2025/Conference — ICML 2025 poster_

### Official Review · Reviewer_Qi3Y · 2025-02-23

**Overall Recommendation:** 4

**Summary:**

The paper "Efficient and Privacy-Preserving Soft Prompt Transfer for LLMs" introduces POST (Privacy Of Soft-prompt Transfer), a framework that enables efficient and privacy-preserving soft prompt tuning and transfer for Large Language Models (LLMs).
Key Contributions:

Privacy-Preserving Soft Prompt Tuning: POST allows users to tune soft prompts locally on a small, distilled model instead of directly on a large LLM, avoiding the need to share private data with external providers.

Knowledge Distillation for Transferability: The LLM provider distills a small version of their model, which the user uses for tuning. The tuned prompt is then transferred back to the large LLM using only a small public dataset.

Differential Privacy Integration: POST incorporates differential privacy (DP) to ensure that sensitive user data remains protected even during prompt tuning.

 Efficient and Effective Transfer: The method significantly reduces computational costs while maintaining strong task performance, outperforming existing soft prompt transfer approaches.

Experiments & Findings:

Evaluations on multiple LLM architectures (RoBERTa, GPT2-XL, LLaMA2-7B) show that POST achieves high-utility prompt transfer while preserving privacy.

It significantly reduces computational overhead compared to direct prompt tuning on large models.

**Claims And Evidence:**

Yes, most claims are well-supported by empirical evidence, though some could benefit from clearer acknowledgment of limitations.

**Essential References Not Discussed:**

The paper provides a thorough review of related work, covering soft prompt tuning, knowledge distillation, and privacy-preserving techniques. It cites key contributions in these areas, including prior work on soft prompt transfer (Su et al., 2022; Wu et al., 2023), differential privacy (Dwork et al., 2006; Duan et al., 2023), and knowledge distillation (Hinton et al., 2015; Sanh et al., 2019).

**Experimental Designs Or Analyses:**

Yes, the experimental design is generally sound. The authors use multiple LLMs, diverse classification and generation datasets, and compare POST against relevant baselines (Zero-Shot Transfer, DP-OPT). The LiRA attack for privacy evaluation and runtime analysis for efficiency are well-executed.

**Methods And Evaluation Criteria:**

Yes, the proposed methods and evaluation criteria are appropriate for the problem. The paper uses knowledge distillation, differential privacy, and public dataset-based transfer, which align well with the goal of efficient and privacy-preserving prompt tuning. The evaluation includes multiple LLMs (RoBERTa, GPT2-XL, LLaMA2-7B) and diverse classification and generation tasks, making the results robust. However, the impact of public dataset choice on privacy and transferability could be further analyzed.

**Other Comments Or Suggestions:**

Section 4.3 (Privacy-Preserving Prompt Transfer): The explanation of the loss function (Equation 3) would be clearer with a brief discussion on the intuition behind the two KL-divergence terms and why they are necessary for effective transfer.

**Other Strengths And Weaknesses:**

Strengths:

1) Innovative Transfer Method: The paper proposes a novel framework, POST, which introduces a structured approach to soft prompt transfer via knowledge distillation. This is a significant technical improvement over prior works that either required private data access or suffered from severe transfer degradation.

2) Efficient Soft Prompt Tuning: By using a distilled smaller model for local tuning, POST significantly reduces the computational cost of optimizing prompts compared to full LLM fine-tuning.

3) Strong Empirical Support: The framework is rigorously tested across multiple model architectures (RoBERTa, GPT-2 XL, LLaMA-2 7B) and five classification plus two open-ended generation tasks, showing strong transfer performance while preserving privacy.


Weaknesses:

1) Limited Theoretical Analysis of Transferability: While the empirical results support the effectiveness of POST, the paper lacks a theoretical discussion on why soft prompts trained on a distilled model remain transferable. The connection between knowledge distillation’s preservation of model decision boundaries and its impact on prompt representation is not analyzed in depth. Including theoretical insights or at least empirical ablations on feature space alignment between the small and large models would strengthen the argument.

2) Prompt Transfer Loss Function Justification: The prompt transfer objective consists of two KL-divergence terms (Equation 3), but there is no principled justification for the chosen balance parameter αα. The weighting scheme appears to be chosen heuristically based on downstream task performance. A more rigorous exploration of how αα interacts with factors such as the student model's generalization error and the zero-shot capabilities of the target LLM is needed.

3) Prompt Transfer Loss Function Is Not Well-Justified: The prompt transfer objective (Equation 3) consists of two KL-divergence terms, but the balance between them is controlled by an empirically chosen hyperparameter αα. There is no discussion on how to select αα beyond empirical tuning, nor any exploration of its effect on convergence or transfer quality.

**Questions For Authors:**

1) How does the choice of public dataset affect the quality of prompt transfer?

    1) The paper mentions that a small public dataset is used for transferring the prompt but does not provide a systematic analysis of how dataset properties (e.g., domain similarity, label space overlap, dataset size) impact transfer success.
    2) Could the method fail if the public dataset is too different from the private task dataset? Have you experimented with mismatched public datasets?

2) Why does the KL-divergence transfer loss (Equation 3) use a fixed weighting factor αα, and how was it selected?

    1) The transfer objective combines two KL-divergence terms with a fixed αα, but there is no theoretical justification for how αα should be set.
    2) Have you explored dynamic or task-adaptive weighting schemes for αα? Could tuning αα differently for various LLMs improve performance?

**Relation To Broader Scientific Literature:**

The paper's contributions relate to broader literature in several key ways:

Soft Prompt Tuning & Transfer: It builds on prior work on soft prompt tuning (Shin et al., 2020; Lester et al., 2021) and addresses the challenge of cross-model prompt transfer, improving on methods that require private data sharing (Su et al., 2022) or suffer from poor performance (Wu et al., 2023).

Privacy-Preserving Adaptation: Unlike prior differential privacy (DP) methods that protect model outputs (Duan et al., 2023), POST ensures privacy during prompt tuning without leaking private data.

Knowledge Distillation for Transfer: It repurposes knowledge distillation (Hinton et al., 2015; Sanh et al., 2019) to enhance soft prompt transfer, rather than just compress models.

Efficiency & Benchmarking: The framework reduces computational costs compared to full fine-tuning and outperforms existing private prompt transfer methods (Hong et al., 2023) across multiple tasks.

**Theoretical Claims:**

The paper does not have any theoretical claims.

---

> ### Author Rebuttal · Authors · 2025-03-31
>
> We thank the Reviewer for the constructive feedback! We address the main points one by one below:
>
> >**A more detailed discussion on public dataset selection risks**
>
> We investigate the effect of the choice of public data in Table 15 in Appendix (section D.1), where we consider both public datasets from similar task domains as well as those from different task domains compared to the private dataset. Our experimental results show that selecting a public dataset with a similar task to the private dataset is helpful for effective prompt transfer.
>
> To limit the privacy leakage and obtain good performance, we showed that only the target task should be considered when selecting the public dataset. In Table 15 (Appendix D.1), we find that the best transfer performance can be achieved with public datasets from the same task family, even if the datasets are not very similar and have, e.g., different numbers of classes. Our framework assumes that the model provider has knowledge of only the task domain, not the user’s actual data. This is a reasonable assumption, as task metadata is often available in practical settings.
> For instance, cloud providers and ML platforms (e.g., Google Vertex AI, OpenAI’s API) typically encourage users to specify the task type or provide task demonstrations (i.e., few-shot learning) when fine-tuning models to optimize performance [A]. Similarly, many AI service providers, such as NVIDIA’s NeMo framework, offer task-specific APIs explicitly designed for applications like chatbots, summarization, and generation, which inherently expose task-level information [B].
>
> >**why soft prompts trained on a distilled model remain transferable？Theoretical insights or at least empirical ablations on feature space alignment between the small and large models**
>
> We conducted an ablation study to investigate the impact of feature alignment. We use a model with the same architecture and initial parameters as our distilled 2-layer LLaMA. Instead of applying knowledge distillation on the BookCorpus dataset, we directly fine-tune this model on BookCorpus to achieve comparable performance to the distilled smaller model. This model has no feature space alignment with the large model. We report this model’s transfer performance.
>  | Dataset | PT on Distilled model | PT on non-distilled model | Transfer with the distilled model | Transfer with non-distilled model |
> |-|-|-|-|-|
> | SST2| 78.78 | 78.24  | 90.02| 85.38|
> | IMDB| 79.95 | 79.40 | 87.29| 82.45|
> | tweet   | 54.12| 54.79| 61.15  | 45.65 |
> | arisetv | 77.92| 72.07| 86.71 | 70.65 |
> | mpqa    | 83.82 | 84.16 | 87.37 | 81.52|
>
> Transfer results for the distilled model are always higher. This indicates that using knowledge distillation to preserve alignment between small model and big model is essential in our designed prompt transfer procedure.
>
> >**How to select $\alpha$ in the prompt transfer loss (Equation 3)? Have you explored dynamic or task-adaptive weighting schemes for $\alpha$? Could tuning $\alpha$ differently for various LLMs improve performance?**
>
> We conducted a thorough ablation study on $\alpha$ (see Table 19), evaluating transfer performance across $\alpha$ values ranging from 0.0 to 1.0. The results indicate that multiple $\alpha$ values lead to good transfer performance, indicating the robustness of our POST to the choice of $\alpha$.
>
> Additionally, we clarify the intuition behind the transfer loss design in Appendix D.5 and propose a heuristic method (Equation 6) for finding an optimal $\alpha$, which supports tuning $\alpha$ differently for different LLMs to improve performance in practice.
>
> >**How dataset properties (e.g., domain similarity, label space overlap, dataset size) impact transfer success?**
>
> We investigated the influence of the domain similarity of the public dataset on the transfer performance (see Appendix D.1). The findings demonstrate that selecting a public dataset with a task domain similar to the private task is helpful for effective prompt transfer.
>
> >**What if the public dataset is too different from the private task dataset?**
>
> Table 15 presents the transfer performance using public datasets from different task families. It is observed that the best transfer performance can be achieved when using same-domain data, while datasets with different tasks can lead to performance degradation.
>
> ---
>
> We provide the reference for all the responses.
>
> **References:**
>
> [A] OpenAI Fine-Tuning Guide – https://platform.openai.com/docs/guides/fine-tuning
>
> [B] NVIDIA NeMo Framework – https://docs.nvidia.com/nemo-framework/user-guide/latest/overview.html
>
> [C] Duan, H., Dziedzic, A., Papernot, N., and Boenisch, F. “Flocks of stochastic parrots: Differentially private prompt learning for large language models.” NeurIPS 2023.

---

### Official Review · Reviewer_n3JH · 2025-03-04

**Overall Recommendation:** 4

**Summary:**

This paper addresses the problem of inefficiency and privacy risks in soft prompt tuning on LLM provider-hosted models. Therefore, the authors propose a prompt transfer framework called POST, that first distill a smaller model to better match the teacher behavior, then prompt tuning on the small model using private data, and finally transfer the prompt to the larger model with a non-private dataset.

**Claims And Evidence:**

Yes the claims made in the submission supported by clear and convincing evidence.

**Essential References Not Discussed:**

No.

**Experimental Designs Or Analyses:**

The experimental settings are generally comprehensive, the proposed POST method is compared with 1. zero-shot performance on private data 2. performance of private data prompt tuning on target model 3. performance of private data prompt tuning on source distilled model and 4. directly transfer the learned prompt to target model that not using public data. POST significantly outperforms the directly transferred prompt and achieves better performance than the distilled source model, demonstrating its practical effectiveness.

**Methods And Evaluation Criteria:**

Yes the proposed three-step prompt transfer method makes sense for the problem of addressing data privacy and prompt tuning efficiency.

**Other Comments Or Suggestions:**

No

**Other Strengths And Weaknesses:**

Strengths:
1. The proposed method is well-motivated
2. The analysis is comprehensive

Weaknesses:
1. The quality of the transferred prompt heavily depends on the distilled model. For complex tasks, where the full target LLM can effectively learn a strong prompt, a weakly tuned prompt on the small model will not transfer well, as the distilled model may fail to capture the necessary task structure.

**Questions For Authors:**

See weakness.

**Relation To Broader Scientific Literature:**

The proposed method sits at the intersection of knowledge distillation, soft prompt tuning and privacy-preserving ML. It improves soft prompt transferability in Wu et al., 2023 and privacy issue in Su et al., 2022.

**Theoretical Claims:**

No applicable.

---

> ### Author Rebuttal · Authors · 2025-03-31
>
> We thank the Reviewer for the positive feedback! We address the main points one by one below:
>
> >**The quality of the transferred prompt heavily depends on the distilled model.**
>
> We fully agree with the Reviewer. The influence of compressed model size (number of layers) on performance and compute time is presented in Table 18 in our paper. And we show the results in the table below for the reviewer’s convenience. Indeed, our findings show that while larger distilled models generally improve performance, they also increase computational costs for the user.
>
> |# layers in distilled version|1 layer|2 layers (paper)|3 layers|
> |-|-|-|-|
> |sst2|84.52|87.73|88.53|
> |imdb|78.01|83.96|83.64 |
> |tweet|50.65|54.55|61.50 |
> |arisetv|53.62|82.73|86.45|
> |Distill time|6h 04min|6h 45min|7h 35min|
>
>
> >**For complex tasks, where the full target LLM can effectively learn a strong prompt, a weakly tuned prompt on the small model will not transfer well, as the distilled model may fail to capture the necessary task structure.**
>
> Beyond investigating the trade-offs of different compression ratios, we also designed a heuristic approach to optimize the $\alpha$ in Equation 6, which balances the first and second loss terms in the transfer loss function (Equation 3). Specifically, when the full target LLM has a strong zero-shot performance or the small model has a weak performance, we set a larger $\alpha$, reducing the reliance on the small model’s behavior while primarily incorporating the directional change induced by the prompt. This adjustment enhances the effectiveness of our POST method in scenarios where the small model has weak performance.
> We perform a wide range of experiments using different $\alpha$ values and compare the best performing $\alpha$ with the one output by our heuristic, as shown in Table 19 in Appendix D.5. For reviewer’s convenience, we show partial results of Table 19, i.e., the performance of Roberta-base, in the following table (transferred accuracy are presented as mean (standard deviation)):
>
> |$\alpha$|sst2|imdb|tweet|arisetv|
> |-|-|-|-|-|
> |heuristic|0.76|0.77|0.14|0.41|
> |0.0|80.54 (1.23)|78.22 (0.55)|57.75 (1.88)|83.41 (0.91)|
> |0.1|81.94 (1.87)|79.20 (2.92)|57.68 (1.95)|83.09 (1.45)|
> |0.2|83.33 (0.78)|80.89 (1.85)|**58.13 (2.16)**|83.62 (0.77)|
> |0.3|84.67 (0.73)|80.57 (1.16)|58.08 (1.15)|83.62 (1.26)|
> |0.4|86.81 (0.46)|79.86 (1.20)|54.77 (1.04)|**83.94 (0.79)**|
> |0.5|88.61 (0.46)|82.16 (1.55)|54.10 (0.74)|80.11 (1.73)|
> |0.6|**88.61 (0.78)**|81.82 (1.06)|52.88 (0.44)|76.57 (0.84)|
> |0.7|87.88 (0.29)|80.89 (1.59)|52.33 (0.20)|71.26 (0.64)|
> |0.8|87.35 (0.24)|**82.65 (0.95)**|51.32 (0.24)|68.68 (0.57)|
> |0.9|86.47 (0.64)|81.52 (1.39)|49.38 (0.55)|63.04 (0.85)|
> |1.0|85.82 (0.85)|80.57 (1.32)|48.42 (1.13)|58.57 (0.24)|
>
> We observe from the results that the heuristic can successfully identify a good range for $\alpha$ and is usually not far off the empirical optimal values.

---

> > ### Comment · Reviewer_n3JH · 2025-04-01
> >
> > Thank you for the additional results. I would like to keep my score as 4.

---

> > > ### Author Response · Authors · 2025-04-02
> > >
> > > We appreciate the Reviewer's positive feedback and for maintaining the high score.

---

### Official Review · Reviewer_f1xX · 2025-03-12

**Overall Recommendation:** 3

**Summary:**

This paper proposes POST (Privacy Of Soft-prompt Transfer), a framework designed to efficiently and privately transfer soft prompts for adapting large language models (LLMs) to private downstream tasks. The core innovation involves locally tuning soft prompts on a small, distilled model derived from a larger LLM, optionally under differential privacy constraints, and subsequently transferring these prompts back to the large LLM via a small public dataset. The method reduces computational requirements and enhances privacy, avoiding the direct sharing of sensitive data with LLM providers. Experimental evaluations using various classification and generation tasks show POST's improvements over existing prompt transfer methods.

**Claims And Evidence:**

The claims in the paper are generally well-supported by empirical results, including evaluations on several datasets (sst2, imdb, tweet, arisetv, mpqa, disaster, trec) across classification and open-ended generation tasks. However, more rigorous exploration into why certain public datasets work better for transfer than others would enhance the clarity of some claims about dataset suitability.

**Essential References Not Discussed:**

n/a

**Experimental Designs Or Analyses:**

The experimental designs and analyses are sound. The authors thoroughly evaluated their methods, including ablations for the impact of the number of public samples, the number of transfer steps, and model compression ratios. However, a clearer explanation of the hyperparameter choices (especially α in the transfer loss) is needed.

**Methods And Evaluation Criteria:**

The proposed methods and evaluation criteria are appropriate and clearly defined for the targeted privacy and efficiency goals. The datasets selected for evaluation (sst2, imdb, etc.) are standard benchmarks, which strengthens the credibility of the results. However, additional clarification or justification for the choice of public datasets is needed.

**Other Comments Or Suggestions:**

An in-depth analysis or case study illustrating practical privacy impacts beyond membership inference attacks would further strengthen the narrative.

**Other Strengths And Weaknesses:**

Strengths:

1. Innovative combination of knowledge distillation, differential privacy, and prompt tuning.

2. Comprehensive experimental validation demonstrating clear benefits over existing methods.

3. Significant practical implications for privacy-sensitive applications of LLMs.

Weaknesses:

1. The choice and generality of the public dataset for prompt transfer are not fully justified.

2. Potential limitations in transfer performance to LLMs with significantly different architectures or training paradigms are not fully explored.

**Questions For Authors:**

1. How robust is POST's performance when the public dataset significantly differs from the private dataset in distribution or domain? Would this impact the transfer accuracy significantly?

2. Can the authors clarify how sensitive the method is to the choice of the α parameter in the prompt transfer loss? How was α tuned in practice?

3. How would the approach scale to very large LLMs (e.g., 70B+ parameters)? Would computational constraints significantly limit the practicality of POST in such scenarios?

**Relation To Broader Scientific Literature:**

The paper positions itself clearly within the broader literature on soft prompts, knowledge distillation, and differential privacy, effectively differentiating itself by combining these aspects into a cohesive framework. The distinction and improvement over previous methods like DP-OPT and zero-shot transfer (ZST) are explicitly demonstrated.

**Theoretical Claims:**

The paper does not contain explicit theoretical claims.

---

> ### Author Rebuttal · Authors · 2025-03-31
>
> We appreciate the Reviewer's insightful comments! Below, we address each comment in detail:
>
> >**More rigorous exploration into why certain public datasets work better for transfer than others; Additional clarification or justification for the choice of public datasets.**
>
> We thoroughly analyzed the effect of public dataset selection in Table 15 in the Appendix (section D.1). Our experimental results demonstrate that public datasets from the same task domain as the private dataset yield better transfer performance than others. This aligns with the intuition that task similarity enhances knowledge transfer. Based on these findings, it is suggested to select a public dataset for a similar task as the private data for effective prompt transfer.
>
> >**A clearer explanation of the hyperparameter choices (especially $\alpha$ in the transfer loss)**
>
> We conducted an ablation study for the parameter $\alpha$ in Table 19 in the Appendix (section D.5).
> We evaluated the transfer performance across various $\alpha$ values, ranging from 0.0 to 1.0. The results indicate that multiple $\alpha$ values lead to good transfer performance.
> Furthermore, we designed a heuristic to find an optimal $\alpha$ as shown in Equation 6 in the Appendix (section D.5). The intuition of the heuristic is that if the target model performs well, it needs to less mimic the behavior of the smaller model, but only incorporate the directional change induced by the prompt, leading to a larger $\alpha$. It can be observed from Table 19 that the designed heuristic can successfully identify a good range for $\alpha$ and is usually not far off the empirical optimal values.
> Thus, for each dataset and model, we selected the $\alpha$ according to the heuristic we proposed in the same appendix section (D.5).
>
> >**Potential limitations in transfer performance to LLMs with significantly different architectures or training paradigms.**
>
> In our setup, the LLM provider distills a small model from the LLM with the aim of guaranteeing good transfer performance. The transfer performance between different architectures is orthogonal to our problem setup.
>
> >**Practical privacy impacts beyond membership inference attacks**
>
> Membership inference attacks are widely recognized as a gold standard for evaluating privacy leakage. Notably, in its most stringent form, our POST framework incorporates DP, which, by definition, ensures that the presence or absence of any individual data point remains indistinguishable and provides theoretical protection against **every possible attack**.
>
> >**Can the authors clarify how sensitive the method is to the choice of the $\alpha$ parameter in the prompt transfer loss? How was $\alpha$ tuned in practice?**
>
> We performed an ablation study on different $\alpha$ values in Table 19, confirming that our approach is robust to a range of $\alpha$-s. Additionally, we designed a heuristic to find an optimal alpha (see Equation 6). The underlying intuition is that if the target model exhibits strong zero-shot performance, it requires less direct imitation of the small model. Instead, it should primarily incorporate the directional changes induced by the prompt, ensuring effective transfer while maintaining the target model’s inherent capabilities.
>
> >**How would the approach scale to very large LLMs (e.g., 70B+ parameters)?**
>
> The LLM providers are assumed to have the computational resources to distill the target models into smaller ones, as they already manage to train such large language models. The overhead on the client end should not depend on the scale of the model used by the LLM provider.

---

### Official Review · Reviewer_SqSB · 2025-03-14

**Overall Recommendation:** 1

**Summary:**

This paper proposes POST (Privacy Of Soft-prompt Transfer), a framework designed to enable efficient and privacy-preserving transfer of soft prompts between LLMs. It has three major steps:
* Knowledge Distillation: The LLM provider first distills a smaller local model from the original large LLM.
* Local Prompt Tuning: The user tunes soft prompts on the small model using their private data (with optional differential privacy for additional protection).
* Privacy-Preserving Prompt Transfer: The LLM provider transfers the tuned prompt back to the large LLM using a small public dataset, avoiding any access to private data.

**Claims And Evidence:**

No obvious wrong claims found.

**Essential References Not Discussed:**

No obvious missing references found.

**Experimental Designs Or Analyses:**

I did check the experiments in the paper. Detailed comments are in "Strengths and Weaknesses" below.

**Methods And Evaluation Criteria:**

Methods are fine for the assumed scenario, but the problem setting itself is unrealistic to me.

**Other Comments Or Suggestions:**

no

**Other Strengths And Weaknesses:**

**Strengths:**
The paper is well-written and easy to follow.

**Weaknesses:**

My major concern is the unrealistic Problem Setting:
* The proposed approach assumes that the LLM provider will release a distilled smaller version of their strong and close-sourced LLM. However, this is impractical in real-world scenarios because the distilled model parameters can inadvertently reveal crucial pretraining details, such as the data weighting of different training corpora [1].
* The approach also requires the LLM provider to execute the proposed soft prompt transfer, which involves selecting a public dataset that is expected to be similar to the private dataset used by the user. This assumption is problematic because, in a proper setting, the LLM provider should not have knowledge of users' private data.

Important models and baselines are also missing in the experiments:
* The experiments are conducted using base models such as Llama2-7B and GPT2-XL. However, real-world LLM services typically deploy instruction-tuned models (e.g., Llama2-7B-Instruct), which exhibit significantly stronger zero-shot problem-solving capabilities. The absence of instruction-tuned models makes the "Full ZS" baseline unconvincing.
* Prompt tuning generally performs worse than full fine-tuning or LoRA fine-tuning while offering only marginal reductions in training cost since it still requires backpropagation through the full model. The key advantage of prompt tuning is its reduced parameter storage requirements. Therefore, a crucial missing baseline is full fine-tuning on the compressed model. Demonstrating superiority only over the prompt-tuned compressed model ("Compressed PT") is not sufficient; the study should compare against a fully fine-tuned small model to justify the proposed approach. For example, if the fully finetuned compressed model is better than the proposed approach, why should the users keep paying for the LLM provider's service?

[1] Detecting Pretraining Data from Large Language Models, ICLR 2024.

**Questions For Authors:**

* What's the expected similarity between the private and public datasets in the proposed framework?
* I understand the soft prompt sent from the private data owner to the LLM provider should definitely have DP guarantees. Otherwise the privacy is not protected at all. Why do you say the DP in this step is just "optional"?

**Relation To Broader Scientific Literature:**

I don't see clear contributions of the paper.

**Theoretical Claims:**

The paper doesn't have theoretical claims.

---

> ### Author Rebuttal · Authors · 2025-03-31
>
> We appreciate the Reviewer's constructive review and address each point as follows:
>
> >**The problem setting itself is unrealistic to me.**
>
> Soft prompts are exposed via public APIs, such as NVIDIA NeMo, as we highlight in the abstract and introduction of our paper.  Could the reviewer kindly clarify specific concerns regarding the “*realism of the problem setting*” in our work?
>
> >**I don't see clear contributions of the paper.**
>
> We enumerated the contributions of our work at the end of the introduction.
>
> >**distilled model parameters can reveal pretraining details [1].**
>
> The problem of protecting pre-training details is orthogonal to our work. However, we do agree that the LLM provider might not want to disclose such details.
>
> To address the reviewer’s comment, we follow the setup from the paper cited by the Reviewer [1] and use mink% to perform Membership Inference Attack on the WikiMIA [1] dataset with the Pythia-2.8B model.  We test 2 models distilled from Pythia-2.8B, each with 3, and 4 layers, respectively. Our results are depicted below and show that the AUC and TPR@1%FPR of the distilled models are much smaller than the base model. This shows that the knowledge distillation in our work can even reduce the pretraining data leakage.
>
> ||Pythia 2.8B|3-layer distilled|4-layer distilled|
> |-|-|-|-|
> |ROC-AUC|0.7103|0.5737|0.5750|
> |TPR@1%FPR|0.0980|0.0196|0.0196|
>
> >**The LLM provider shouldn't have knowledge of users' private data.**
>
> Our framework assumes that the model provider has knowledge of only the task domain, not the user’s actual data. This is a reasonable assumption, as task metadata is often available in practical settings. For instance, cloud providers and ML platforms (e.g., Google Vertex AI, OpenAI) typically encourage users to specify the task type or provide task demonstrations (i.e., few-shot learning) to fine-tune models [A]. Similarly, many AI service providers, such as NVIDIA’s NeMo framework, offer task-specific APIs explicitly designed for applications like chatbots and summarization, which inherently expose task-level information [B].
> Importantly, our definition of *privacy leakage* follows DP and is measured at the individual datapoint level. Our approach follows the practical setups. We also offer a general statement regarding the privacy-utility trade-off at the end of our response.
>
> >**Additional baseline 1: Zero-shot performance of instruction-tuned models**
>
> We appreciate the suggestion and have included the zero-shot performance of instruction-tuned Llama2-7b-chat in the following table:
> |SST2|IMDB|Tweet|arisetv|mpqa|
> |-|-|-|-|-|
> |81.42|85.01|44.55|77.29|54.37|
>
> The instruction-tuned model has very close zero-shot performance to the non-instruction-tuned model. This is because we convert the problem of classification to a prefix infilling problem for a non-instruction-tuned model and aggregate the results with multiple ground truth text labels, see Table 7 in the Appendix. This enhances the robustness of our evaluation and allows the non-instruction-tuned model to effectively handle classification tasks.
>
> >**Advantage of prompt tuning**
>
> Prompt tuning offers advantages for LLM providers as it enables batch processing, allowing multiple tasks (from different users) to be handled in a single inference pass with prompts. In contrast, LoRA or fine-tuned models require separate inference passes through the model, significantly increasing the computational costs, and even more importantly, separate models per task [C]. The efficiency gain of soft-prompts provides a strong incentive for the LLM providers to adopt our framework.
>
> >**Additional baseline 2: Full fine-tuning on the compressed model**
>
> We have conducted an additional experiment evaluating full fine-tuning on the compressed model:
> ||SST2|IMDB|tweet|arisetv|mpqa|
> |-|-|-|-|-|-|
> |Llama2-7b-compressed|83.02|83.39|59.21|78.41|85.23|
>
> Full fine-tuning on the compressed model indeed outperforms prompt tuning on the compressed model, however, it still underperforms POST transfer. This is because the compressed model, although it maintains some similarity to the teacher model, is still not powerful enough.
>
> >**expected similarity between the private and public datasets**
>
> Table 15 in Appendix D.1 investigates the impact of public dataset selection. We categorize datasets into three groups based on task similarity and observe that public datasets with tasks similar to the private dataset yield better transfer performance. Thus, it is expected that the public dataset is from the similar task domain as the private dataset in our framework.
>
> >**Why the DP is “optional”?**
>
> Our framework is designed to be modular, allowing private data owners to balance privacy and utility trade-offs based on their specific requirements. DP is an optional component that can be integrated when strong privacy guarantees are needed, but its inclusion depends on the user’s trade-off preferences.
>
> ---
>
> References included in answer to Reviewer Qi3Y.

---

> > ### Comment · Reviewer_SqSB · 2025-04-05
> >
> > Thank you to the authors for the additional experiments — they help alleviate some of my concerns regarding unfair comparisons in the evaluation.
> >
> > However, I still don't feel that my major concerns have been well addressed:
> >
> > > Could the reviewer kindly clarify specific concerns regarding the “realism of the problem setting” in our work?
> > > Soft prompts are exposed via public APIs, such as NVIDIA NeMo.
> >
> > My concerns regarding the problem setting are described in the "Weaknesses" section. Your framework assumes that the parameters of a distilled version of a closed-source LLM can be released to users for prompt tuning, which is different from the soft prompt APIs mentioned here.
> >
> >
> > > The problem of protecting pre-training details is orthogonal to our work.
> >
> > I am not convinced by this point. On the contrary, it should be one of the most critical aspects to validate in advance. Your work proposes an LLM service based on a closed-source model, whose pretraining costs exceed hundreds of millions of dollars and whose training details are highly confidential. Therefore, releasing a distilled version of such a model would be a highly risky action — *yet this highly risky action becomes a fundamental prerequisite for your proposed framework*. To validate your approach, it is essential to thoroughly demonstrate that the distillation is secure against possible attacks, theoretically or empirically, such as "how large can the distilled model be while still guaranteeing protection?", etc. The single attack example you added is not sufficient, though I appreciate the effort.
> >
> >
> > > Cloud providers and ML platforms typically encourage users to specify the task type or provide task demonstrations (i.e., few-shot learning) to fine-tune models.
> >
> > "Whether the service provider should know the user's private task information" might be debatable. However, the two examples you cite — OpenAI [A] and NeMo [B] — do not actually guarantee data privacy for users, based on the links you provided.
> >
> >
> > Overall, the assumptions underlying your proposed privacy-preserving LLM service framework — particularly regarding distillation — are not thoroughly justified. As such, it is difficult to envision how the proposed framework could be practically and securely implemented.

---

> > > ### Author Response · Authors · 2025-04-05
> > >
> > > >**Your framework assumes that the parameters of a distilled version of a closed-source LLM can be released to users for prompt tuning (...) To validate your approach, it is essential to thoroughly demonstrate that the distillation is secure against possible attacks, theoretically or empirically, such as "how large can the distilled model be while still guaranteeing protection?", etc. The single attack example you added is not sufficient, though I appreciate the effort.**
> > >
> > > The additional experiment that we performed for the Reviewer runs a state-of-the-art MIA attack against the largest distilled models used in our work (3 and 4 layers), i.e., the ones with the highest potential of leakage. As we highlight in the paper, POST assumes an aggressive distillation to extremely small models, as this enables efficient local prompt tuning on user-hardware, as we show in Table 13 and Table 18. The question on “how large can the distilled model be while still guaranteeing protection”, therefore, does not capture the core of our method.
> > >
> > > >**Overall, the assumptions underlying your proposed privacy-preserving LLM service framework — particularly regarding distillation — are not thoroughly justified. As such, it is difficult to envision how the proposed framework could be practically and securely implemented.**
> > >
> > > To deploy POST, a concerned LLM provider can perform a multitude of state-of-the-art attacks against the distilled model to assess its risk before leakage. If the attacks yield a risk that the LLM provider deems intolerable, they can further compress the model, thereby, reducing leakage.
> > >
> > > >**"Whether the service provider should know the user's private task information" might be debatable. However, the two examples you cite — OpenAI [A] and NeMo [B] — do not actually guarantee data privacy for users, based on the links you provided.**
> > >
> > > We agree with the Reviewer that existing APIs do not guarantee user privacy. Our proposed POST framework offers a solution to this problem. Therefore, especially when executed with DP, it provides the formal guarantee that no individual user data points will not leak more to the LLM provider than specified by the DP parameters.

---

### Decision · Program_Chairs · 2025-05-01

**Decision:**

Accept (poster)

**Comment:**

The paper submission got initially a Reject, two Accept, and one Weak Accept ratings. While reading the discussion in the rebuttal phase, I acknowledge that the problem setting can be challenging to achieve (Reviewer SqSB). However, as other Reviewers state there are some innovations: combination of knowledge distillation, differential privacy, and prompt tuning (Reviewers f1xX and Qi3Y). After an internal discussion w/ the Reviewers, all of them kept their ratings. Given the innovations, I can recommend the acceptance of the paper. However, I think the submission has to discuss the limitations of the proposed approach raised by Reviewer SqSB and include the additional results posted in the rebuttal discussion.